# Slow vibrational relaxation drives ultrafast formation of photoexcited polaron pair states in glycolated conjugated polymers

Katia Pagano [1,9], Jin Gwan Kim[2,9], Joel Luke [1], Ellasia Tan[1], Katherine Stewart[1], Igor V. Sazanovich [3], Gabriel Karras[3], Hristo Ivov Gonev [4], Adam V. Marsh[5], Na Yeong Kim[2], Sooncheol Kwon[6], Young Yong Kim [7], M. Isabel Alonso [8], Bernhard Dörling [8], Mariano Campoy-Quiles [8], Anthony W. Parker [3], Tracey M. Clarke [4], Yun-Hi Kim [2] ✉ & Ji-Seon Kim [1] ✉

Glycol sidechains are often used to enhance the performance of organic photoconversion and electrochemical devices. Herein, we study their effects on electronic states and electronic properties. We find that polymer glycolation not only induces more disordered packing, but also results in a higher reorganisation energy due to more localised $\pi$-electron density. Transient absorption spectroscopy and femtosecond stimulated Raman spectroscopy are utilised to monitor the structural relaxation dynamics coupled to the excited state formation upon photoexcitation. Singlet excitons are initially formed, followed by polaron pair formation. The associated structural relaxation slows down in glycolated polymers (5 ps vs. 1.25 ps for alkylated), consistent with larger reorganisation energy. This slower vibrational relaxation is found to drive ultrafast formation of the polaron pair state (5 ps vs. 10 ps for alkylated). These results provide key experimental evidence demonstrating the impact of molecular structure on electronic state formation driven by strong vibrational coupling.

The demand for high-performance organic semiconducting materials has driven the development of design rules to tailor organic semiconductor (OSC) properties for specific device applications, including organic photovoltaics (OPVs)[1], organic field-effect transistors (OFETs)[2] and organic light-emitting diodes (OLEDs)[3]. OSCs provide many processing advantages for the manufacture of low-cost, scalable and high-performance organic optoelectronic devices. OSCs are, however, intrinsically limited by their low dielectric constants, which arise due to

the fact that OSCs comprise a conjugated system of $\pi$-bonds[4,5]. This also causes OSCs to exhibit large inherent disorder and electronic wavefunctions tend to be localised to single molecules or part of a molecule[6]. As a result, there is strong electron-phonon coupling in OSCs, where electronic transitions are strongly coupled to molecular lattice vibrations (phonons). The energy required for the molecular lattice distortion is known as the reorganisation energy ($\lambda$) and it has two components: the inner $\lambda$, which quantifies the energy required for

[1]Department of Physics and Centre for Processable Electronics, Imperial College London, London SW7 2AZ, UK. [2]Department of Chemistry and Research Institute of Molecular Alchemy (RIMA) Gyeongsang National University Jinju, Gyeongnam 660–701, Republic of Korea. [3]Central Laser Facility, Research Complex at Harwell, STFC Rutherford Appleton Laboratory, Didcot, Oxfordshire OX11 0QX, UK. [4]Department of Chemistry, University College London, Christopher Ingold Building, London WC1H 0AJ, UK. [5]Physical Science and Engineering Division, KAUST Solar Center (KSC), King Abdullah University of Science and Technology (KAUST), Thuwal 23955-6900, Saudi Arabia. [6]Department of Energy and Materials Engineering, Dongguk University-Seoul, Seoul 04620, Republic of Korea. [7]Beamline Division, Pohang Accelerator Laboratory, Pohang University of Science and Technology, Pohang 37673, Republic of Korea. [8]Department of Nanostructured Materials, Institut de Ciència de Materials de Barcelona, ICMAB-CSIC, E-08193 Bellaterra, Spain. [9]These authors contributed equally: Katia Pagano, Jin Gwan Kim. ✉e-mail: ykim@gnu.ac.uk; ji-seon.kim@imperial.ac.uk

the intra-molecular conformational changes ($\pi$-electron redistribution and bond length changes) to occur due to charge formation or electronic excitation and the outer $\lambda$, which quantifies the changes in charge distribution and nuclear positions when the surrounding molecular lattice distorts to accommodate the charge or excited state (ES). $\lambda$ introduces an intrinsic energy barrier to ES and charge formation, and hence charge transport. These properties lead to what is described as hopping transport, where charges are transferred from one molecule to another via the overlapping $\pi$-orbitals[7], limiting the charge carrier mobilities of OSCs[8].

Many design rules have been implemented to reduce $\lambda$ values and enhance OSC device performance. These include enhancing backbone planarity[9], creating long-range order[10] and minimising $\pi$-$\pi$ stacking distances[11]. Sidechain engineering is another design strategy used to modify the physical, electrical and optical properties of conjugated polymers[12]. Incorporation of oligo(ethylene glycol) sidechains into the conjugated backbone has attracted increasing interest for application in a range of organic electronic devices. Compared to alkyl sidechains, glycol sidechains are hydrophilic, more polar and more flexible[13]. In organic electrochemical transistors (OECTs), these properties facilitate ion penetration and result in high-current accumulation mode devices with high transconductance and sharp subthreshold switching[12,14]. Recent work has also highlighted the importance of glycol positioning relative to the conjugated backbone as well as glycol sidechain spacing along the backbone in enhancing OECT transconductance[15]. In organic thermoelectric devices, glycol sidechains have been shown to improve polymer miscibility with dopants[16,17]. Incorporation of glycol sidechains into the conjugated backbone has also been shown to modify polymer optoelectronic properties beneficially for applications in solid-state devices, such as OPVs and OFETs[18]. It has been reported that incorporation of the glycol sidechains can enable closer $\pi$-$\pi$ stacking distances, due to the increased flexibility of the chains and smaller steric hindrance. This in turn enhances the charge-carrier mobility of the polymer[19,20]. The high polarity and flexibility of the glycol sidechains leads to a frequency-dependent enhanced dielectric constant[21], which has attracted interest as a possible mechanism to reduce exciton binding energies in OPVs[4]. The incorporation of glycol sidechains into donor and acceptor materials, or donor-acceptor type non-fullerene acceptors (NFAs) and copolymers typically used in OPVs, however, lags the development of glycolated homopolymers. This is because the benefit of an enhanced dielectric constant, which can improve charge generation, can be accompanied by poorer charge transport properties. For photovoltaics, charge generation, transport and recombination processes are equally important and all need to be optimised for good device performance. Therefore, as photovoltaic applications have been the driving force for the development of glycolated donor, acceptor and donor-acceptor type materials, the need for good charge transport properties in addition to enhanced charge generation has meant that their development is slow. In addition to this, the complicated synthesis process of incorporating glycol sidechains into the donor and acceptor materials requires many more steps than with alkyl sidechains and is often not possible. This has further limited the development of glycolated OPV materials.

While the incorporation of glycol sidechains into conjugated polymer backbones and the impact on device performance has been studied widely, fundamental understanding of the relationship between the polymer molecular structure, $\lambda$ and ES or charge formation is still lacking. Studies relating electron-phonon coupling strength to molecular structure can provide essential information for strategies to minimise $\lambda$ or undesirable decay pathways and can hence facilitate the design of high-performance OSCs for specific device applications. One technique that has been utilised to study the dynamics of excited state formation is femtosecond stimulated Raman spectroscopy (FSRS)[22-25]. FSRS is an ultrafast nonlinear optical technique used to

obtain time-resolved vibrational spectra on timescales of less than 100 fs. FSRS enables the structural dynamics of ultrafast photophysical processes in, for example, conjugated polymers to be characterised and observation of the evolution of excited state structures on sub-picosecond timescales. This provides fundamental understanding of structural relaxation dynamics following excited state formation in the conjugated polymers. Such studies are important to develop understanding of structure-property relationships and hence facilitate the development of molecular design rules for various electronic device applications.

Herein, we investigate the dynamics of ES formation in a series of cyclopentadithiophene (CPDT) polymers, which have varying proportions of monomer units with glycol sidechains. We utilise FSRS to probe the conformational changes induced by photoexcitation in the polymers with the aim of investigating how the glycol sidechains impact electron-phonon coupling and $\lambda$ of the polymers. Using FSRS we define the timescale for structural relaxation of the polymers' backbone following photoexcitation, specific molecular unit(s)' involvement in excited-state (de)localisation, and the impact of glycol sidechains on ES formation dynamics. In addition, transient absorption spectroscopy (TAS) and Raman spectroscopy, are utilised to elucidate the impact of glycol sidechains on electron-phonon coupling and ES formation. Increasing glycol sidechain content is found to alter the ground state optoelectronic properties of the CPDT polymer, with the intrinsic $\pi$-electron density becoming more localised within the CPDT unit. Such localization of ground state $\pi$-electron density is found to induce a larger $\lambda$ upon ES formation. TAS shows that the nature of the ES upon photoexcitation does not vary with glycol content, with singlet excitons forming and decaying to form polaron pairs (PPs) on picosecond timescales, but that the PP signal grows in faster in the glycolated polymer. FSRS reveals that this is due to stronger vibrational coupling in the glycolated polymer facilitating PP formation, which arises due to its slower structural relaxation following ES formation and a larger degree of vibrational cooling occurring on the ES potential energy surface as it structurally relaxes. These findings provide deeper understanding of and insight into the intrinsic properties of glycol sidechains and how these impact ES formation and decay, which will be beneficial for their implementation in the range of organic electronic devices outlined above. In particular, the role of hot excitonic states on charge generation, which is critical for photoconversion efficiency but about which very little is known in the literature, can be identified. By slowing down vibrational cooling (slower structural relaxation) from the hot excitonic states, stronger vibrational coupling is achieved, which leads to ultrafast PP state formation.

## Results

### Absorption and structural properties

The chemical structures of the CPDT polymer series are shown in Fig. 1a. These polymers are synthesised, and their synthesis and chemical characterisation are described in detail in Suppl. Methods and Suppl. Note 1. Each polymer in the series has an identical backbone, consisting of cyclopentadithiophene units, but varying glycol sidechain content. The glycol content percentage of the polymers is determined by the ratio of alkylated and glycolated CPDT monomer units, with g0% comprising purely alkylated monomers and g100% comprising glycolated monomers. Note that the g50% polymer is an alternating copolymer, whereas the g25% polymer is a random copolymer in which glycolated monomers are not adjacent.

We first investigate how varying the glycol sidechain content impacts the ground-state (GS) optoelectronic properties of the polymers. The UV-vis spectra of the CPDT polymers are shown in Fig. 1b. For all polymers, the GS $\pi$-$\pi$* transition band is in the range 450–750 nm. For the g0%, and g25% polymers, the $\pi$-$\pi$* band has a maximum absorption at 632 nm and 637 nm, respectively, assigned to

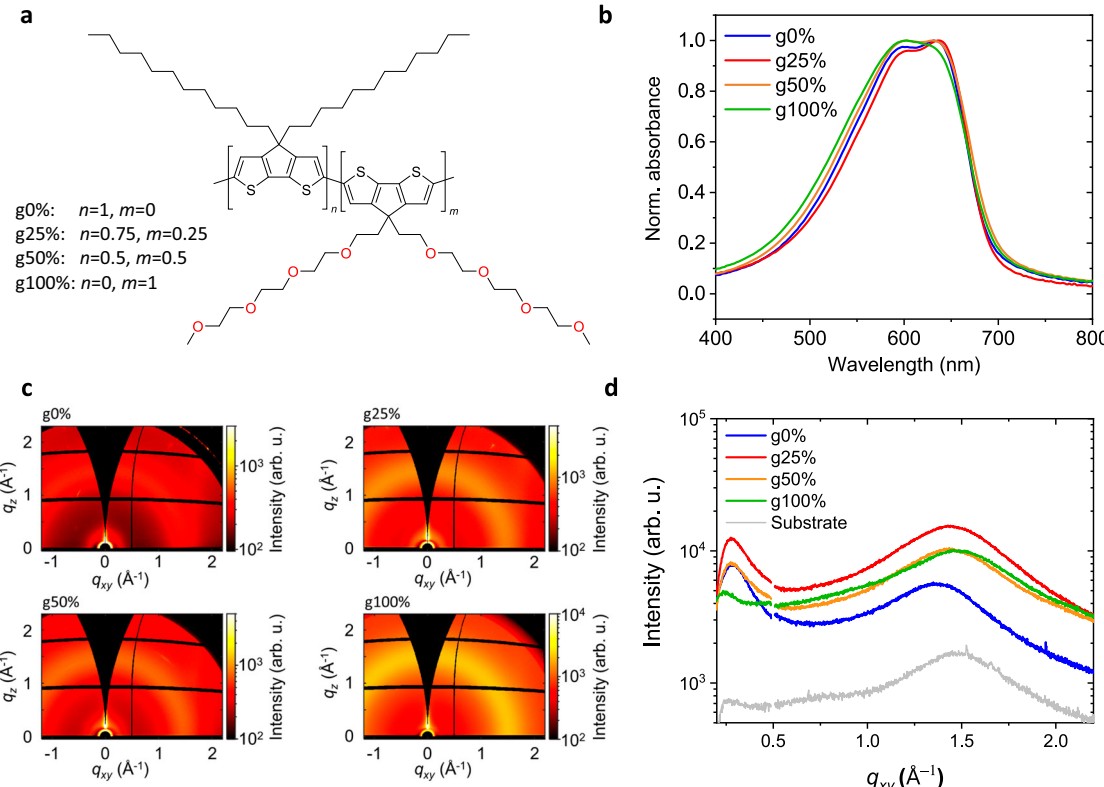

**Fig. 1 | Thin film characterisation. a** General chemical structure of the CPDT polymers. The different glycol content polymers are created by varying the values of *n* and *m* from 0 to 1. **b** Normalised thin-film UV-vis spectra of the CPDT polymer series. **c** 2D GIWAXS patterns of the CPDT polymer series with diffractograms in the (**d**) in-plane ($q_{xy}$) direction. Source data are provided as a Source Data file.

the 0-0 vibronic transition and the 0-1 vibronic shoulder at 597 nm and 600 nm[26]. Increasing the glycol sidechain content above g25%, we see a loss of these vibronic features, a decrease in the ratio of the 0-0 and 0-1 vibronic transition intensities ($A_{0-0}/A_{0-1}$) and a blueshift in the GS absorption maximum. These changes imply that increasing glycol sidechain content induces more disordered packing, hence reducing the crystallinity of the polymer films[27,28]. Although many studies show that replacing alkyl sidechains with more flexible glycol sidechains improves polymer packing[13,19,29], there are others that observe, as we do here, a deterioration in packing order and crystallinity[30,31].

The polymer thin-film packing was also probed with grazing incidence wide-angle X-ray scattering (GIWAXS). Figure 1c shows the 2D GIWAXS patterns for the CPDT polymer series and Fig. 1d shows the scattering profile across the in-plane ($q_{xy}$) axis. The scattering profile across the out-of-plane ($q_z$) axis is shown in Suppl. Fig. 29. The intensity of the lamellar stacking peak is higher in intensity in the $q_z$ than the $q_{xy}$ axis for all polymers, indicating a mostly edge-on packing relative to the substrate. However, lamellar and π-stacking peaks are present in both planes, which suggests that packing orientation in the polymer films is bimodal. Suppl. Table 3 shows the extracted *q* and *d*-spacing values for the in-plane and out-of-plane axes. Increasing glycol sidechain content from g0% to g100% reduces the *d*-spacing of the in-plane π-stacking by 0.3 Å, suggesting that the glycol sidechains induce closer π-π stacking, which is consistent with the studies mentioned previously[13,19,29]. Despite this, the g100% film has a lower intensity of the $q_{xy}^{100}$ lamellar peak and larger lamellar stacking distance compared to the g0% film. So, we conclude that while increasing glycol sidechain content does induce closer π-π stacking, it also increases lamellar stacking distance, resulting in an overall reduction in crystallinity and increase in disorder, consistent with the absorbance data in Fig. 1b.

## Reorganisation energy calculations

To investigate the impact of glycol sidechains on electron-phonon coupling and ES formation in the CPDT polymer series upon photo-excitation the *λ* of the polymers was estimated using simulations and solvatochromic optical measurements. As outlined in the introduction, *λ* is the energy required for the molecular lattice to distort upon ES formation and decay. In our calculations we consider *λ* upon formation of a singlet ES ($S_1$) from the GS. Singlet ES formation can be modelled using the potential energy surfaces (PES) of the neutral and ES molecules, shown in Fig. 2a. In accordance with the Franck-Condon (F-C) principle and Born-Oppenheimer approximation, the atomic nuclei remain stationary during an electronic transition, resulting in a vertical transition (F-C transition) from the ground ($S_0$) to excited ($S_1$) state. If an offset exists between the equilibrium geometries of the $S_0$ and $S_1$ state, the F-C transition results in excitation to a vibrationally ES on the $S_1$ PES. Following the electronic excitation, the molecule will undergo rapid intramolecular vibrational energy redistribution (IVR), in response to changes in electron distribution within the excited molecule, to the minimum of the $S_1$ PES[32]. The molecule can then return to the $S_0$ state via a similar vertical electronic emissive transition (i.e., fluorescence), followed by ultrafast IVR to the minimum of the $S_0$ PES, or via non-radiative internal conversion, where there is a horizontal transition to an isoenergetic vibrational energy level of $S_0$ followed by IVR.

The energy change associated with nuclear relaxations to the minima of the ground and excited-state PESs is the intramolecular (inner) *λ*, $\lambda_i$[33–35]:

$$\lambda_i = \lambda_1 + \lambda_2 \tag{1}$$

Where $\lambda_1$ and $\lambda_2$ are the excited-state and ground-state reorganisation energies, respectively. The PES equilibrium position offset, $\Delta Q$,

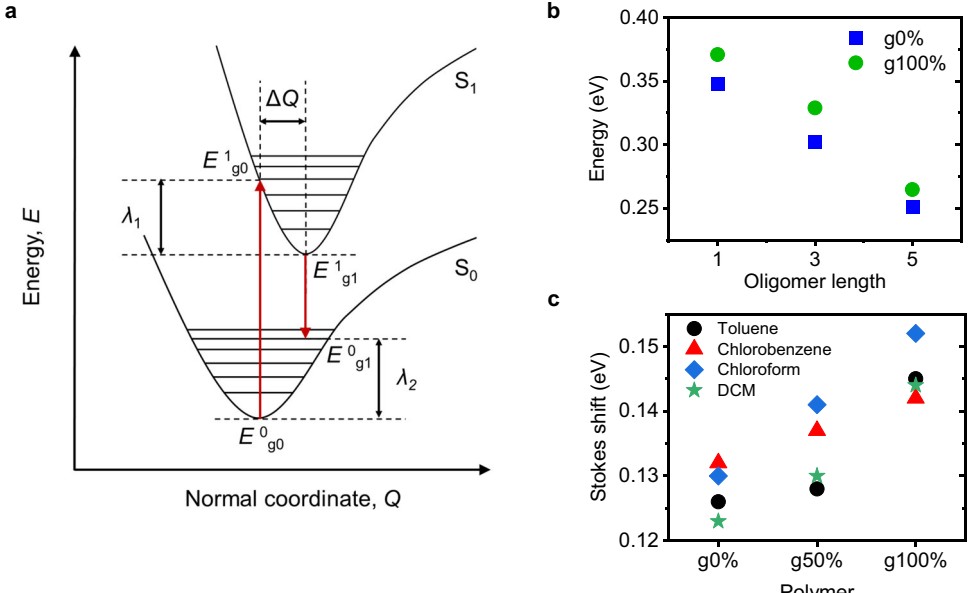

**Fig. 2 | Simulations and solvatochromic data. a** Schematic of the PESs for the ground ($S_0$) and excited ($S_1$) states showing the vertical transitions that occur during the electronic transitions, the PES minima displacement ($\Delta Q$), and the reorganisation energies associated with vibrational relaxation to the PES minimum after an electronic transition occurs. $E^0_{g0}$ is the energy of the ground state molecule at the optimised geometry of the ground state molecule, $E^1_{g0}$ is the energy of the excited-state molecule at the optimised geometry of the ground state molecule, $E^1_{g1}$ is the energy of the ES molecule at the optimised geometry of the ES molecule and $E^0_{g1}$ is the energy of the ground state molecule at the optimised geometry of the ES molecule. $\lambda_1$ and $\lambda_2$ are the reorganisation energies associated with IVR on the $S_1$ and $S_0$ PESs, respectively. **b** DFT-calculated singlet $\lambda$ values for a monomer, trimer and pentamer of the g0% (blue squares) and g100% (green circles) polymers. **c** Stokes shift values taken as the energy difference between the positions of the solution absorption and PL spectra maxima. Source data are provided as a Source Data file.

determines the strength of overlap of the ground and electronically ES vibrational wavefunctions and hence the degree of vibronic coupling to a given vibrational state.

We used Density functional theory (DFT) and time-dependent DFT (TD-DFT) to calculate the $\lambda$ of g0% and g100% oligomers upon singlet exciton formation. DFT calculations simulate isolated molecules in the gas phase, so $\lambda$ values obtained are estimates for the inner $\lambda$. Figure 2b shows the calculated $\lambda$ values for a monomer, trimer and pentamer of the g0% and g100% polymers. $\lambda$ decreases with increasing oligomer length. This is due to the ES wavefunction being able to delocalise further along the conjugated backbone, meaning that smaller conformational changes occur in the backbone upon formation of the $S_1$ state[36]. Comparing the g0% and g100% polymers, the g100% polymer has a larger ($\approx 20$ meV) $\lambda$ for all oligomer lengths. Average dihedral angles of the simulated GS oligomers, Suppl. Fig. 30, show that the g100% oligomers are more twisted, which can induce more localisation of the $\pi$-electron density within the CPDT unit and larger degrees of lattice reorganisation upon ES formation, which is consistent with Raman and FSRS data presented and discussed later. Formation of the ES acts to planarize the oligomer, so an initially more twisted oligomer will undergo larger changes in dihedral angle to accommodate the ES, which correlates with a larger reorganisation energy[1,37,38].

To further investigate the effect of the glycol sidechains on $\lambda$ and provide experimental support for the simulations, solvatochromic measurements were performed to determine Stokes shift values for the g0%, g50%, and g100% polymers in solution. Stokes shift is defined as the energy difference between the absorption and emission maxima[39,40]. From Fig. 2a, it is clear that a larger Stokes shift correlates with a larger $\Delta Q$ and hence a larger $\lambda$[41]. Fig. 2c shows the experimental Stokes shifts of the polymers from the spectra shown in Suppl. Figs. 31, 32. Absorbance and PL spectra were obtained for the polymers in solution to eliminate intermolecular packing effects and the experiments were repeated in solvents of different polarities. The spectra maxima for each polymer and solvent are extracted and shown in Suppl. Table 4. For each solvent, the Stokes shift increases with increasing glycol sidechain content, reflecting an increasing value of $\Delta Q$, and hence $\lambda$, with increasing glycol content, consistent with simulated results.

We also expect there to be intermolecular contributions to the total $\lambda$, i.e., the outer $\lambda$ component. These will arise in the thin film due to differences in morphology and their effects will not be seen in the DFT simulations nor the solvatochromic measurements. From the absorbance data (Fig. 1b), it was found that g100% thin films have more disordered packing and reduced crystallinity compared to the g0% films, which in turn causes larger degrees of molecular structure change upon ES formation[36]. Shorter effective conjugation lengths, hence larger $\lambda$ values, will hinder both intramolecular and intermolecular charge transport. Non-adiabatic Marcus theory states that the inter-molecular charge transfer rate ($k_{if}$) is partially determined by the $\lambda$[42], that is, a larger $\lambda$ will reduce the rate of electron transfer. Consequently, rapid ES relaxation to the ground state is energetically favourable over electron transfer in, for example, exciton separation. The g100% therefore demonstrates both higher inner and outer reorganisation energies than its alkylated counterpart.

## Steady-state Raman spectroscopy

To investigate the effect of the glycol sidechains on the $\pi$-electron distribution along the GS conjugated backbone, steady-state Raman spectroscopy was carried out, which is a vibrational spectroscopy technique that allows us to probe the molecular structure and conformation of conjugated polymers[43]. By comparing the presence and intensities of the Raman peaks between the CPDT polymers, we are able to comment on the nature of $\pi$-electron distribution along the conjugated backbone and how this varies with glycol sidechain content[44]. Figure 3a shows the Raman peak assignments as determined

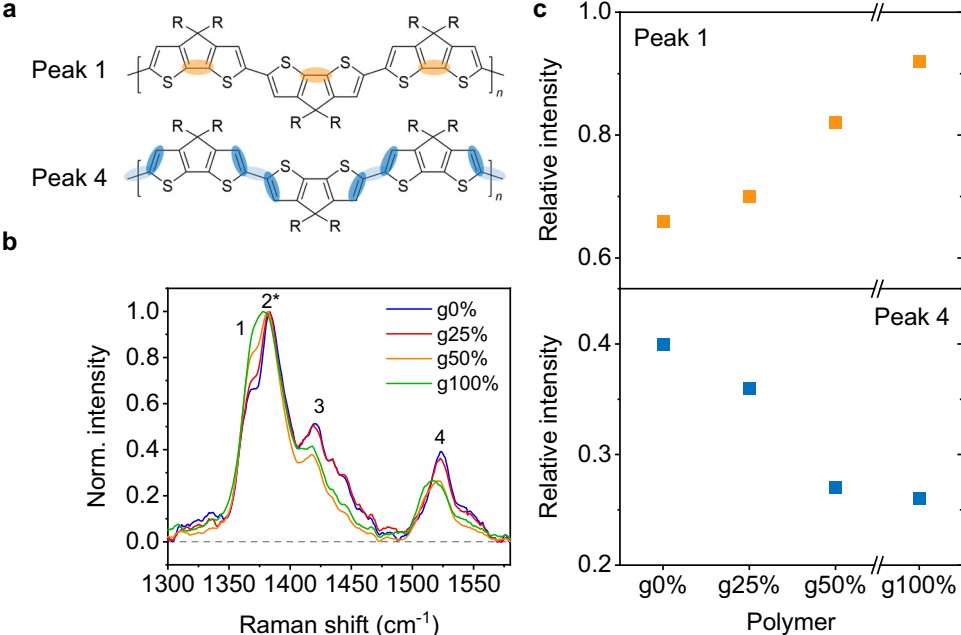

**Fig. 3 | Ground-state Raman spectra. a** Peak assignments of the two vibrational modes of interest. Peak 1, at around 1370 cm⁻¹, corresponds to the C-C intra mode on the cyclopentane ring (shaded orange). Peak 4, at around 1520 cm⁻¹, corresponds to the C = C intra thiophene mode, and the neighbouring C-C inter mode between the CPDT units (shaded blue). **b** Raman spectra for the CPDT polymers, normalised to the 1380 cm⁻¹ mode peak 2*, thin films on prefabricated Fraunhofer OFET substrates, using 785 nm excitation wavelength. **c** Relative peak intensities of peaks 1 and 4 for the CPDT polymers. Source data are provided as a Source Data file.

from DFT simulations, shown in Suppl. Fig. 33 and Suppl. Table 5. Peak assignments are the same for each CPDT polymer. Peak 1 at 1370 cm⁻¹, the shoulder of the main Raman peak at around 1380 cm⁻¹, is assigned to the C-C intra-ring mode on the central cyclopentane ring of the CPDT unit. Peak 3 at around 1420 cm⁻¹ is assigned to the C-C intra-ring mode on the thiophene rings. Peak 4 at around 1520 cm⁻¹ is assigned to the outer C = C mode on the thiophene rings and the neighbouring C-C inter-ring mode between the CPDT units. Figure 3b shows the thin film Raman spectra for the CPDT polymer series with different glycol sidechain content normalised to the most intense 1380 cm⁻¹ peak, peak 2, which is assigned to the C = C modes along the entire backbone coupled with the C-C intra-ring mode on the central cyclopentane ring of the CPDT unit. The relative peak intensities of peaks 1 and 4 are extracted and shown in Fig. 3c. As glycol content is increased, the relative intensity of peak 1 increases while that of peak 4 decreases. This suggests that replacing alkyl sidechains with glycol sidechains induces a more localised π-electron density in the central cyclopentane ring within the CPDT unit, with less inter-unit delocalisation. In addition to the relative peak intensity changes seen in the spectra, peak 4 shows a slight downshift for glycol contents above g25%. This is indicative of an increase in bond length, which arises due to a loss of π-electron density from the thiophene C = C intra and C-C inter bonds as it becomes more localised to the centre of the CPDT unit. A slight broadening of peak 4 can also be seen, which is reflective of the more disordered packing structure in the higher glycol content films.

A more localised π-electron density within the CPDT unit in the g100% polymer is consistent with the higher dihedral angles seen in the DFT simulations (Suppl. Fig. 30). To further support this, DFT was used to simulate the electrostatic potential distribution (ESP) according to the Merz-Kollman (MK) scheme[45] along the backbone of ground state g0% and g100% trimers (Suppl. Fig. 34 and Suppl. Note 2). The results show that compared to the g0% trimer, the g100% trimer has less π-electron density localised to the outer thiophene rings, and more π-electron density localised to the central cyclopentane ring. This is consistent with the experimental results shown in Fig. 3 and

supports the conclusion of the π-electron density being more localised to the centre of the CPDT unit in the g100% polymer.

The differences between the steady-state Raman spectra observed experimentally in thin films here are not visible in the DFT simulated spectra of isolated CPDT trimers with different glycol sidechain content (Suppl. Fig. 33), suggesting that the differences induced by the glycol sidechains are mostly originated from intermolecular interactions and solid-state packing. The absorbance and GIWAXS data show that higher glycol content results in more disordered packing and a reduction in crystallinity, which has been shown to cause localisation of π-electron density in the backbone[36]. Interestingly, these Raman spectra indicate that in thin films the π-electron distribution along the conjugated backbone can be modulated by varying the glycol sidechain content, without any change to the structure of the backbone itself. The more localised nature of the π-electron density in the ground state of the g100% polymer will strongly affect the dynamics of ES formation, as will be discussed below.

### Excited-state dynamics probed with TAS and FSRS

The characterisation techniques presented so far only provide information on the steady-state properties of the CPDT polymers and how these change with increasing glycol sidechain content. However, to understand how the ES formation dynamics are affected by glycol sidechains, time-resolved techniques are essential. TAS is a pump-probe spectroscopic technique used to probe the nature and lifetimes of the ESs generated in a material upon photoexcitation via an actinic pump and shows how the absorption and hence populations of the GS and ESs change over time. While TAS measurements provide information on ES nature and lifetime, it provides little information on the structural relaxation dynamics associated with the ES formation and evolution. For this, we use FSRS, which enables monitoring of the vibrational modes coupled to the electronic transitions observed in the TAS measurements. In particular FSRS enables probing of the benzoidal to quinoidal structural changes that are typically induced in conjugated polymers upon ES formation[36]. Spectroscopically, the

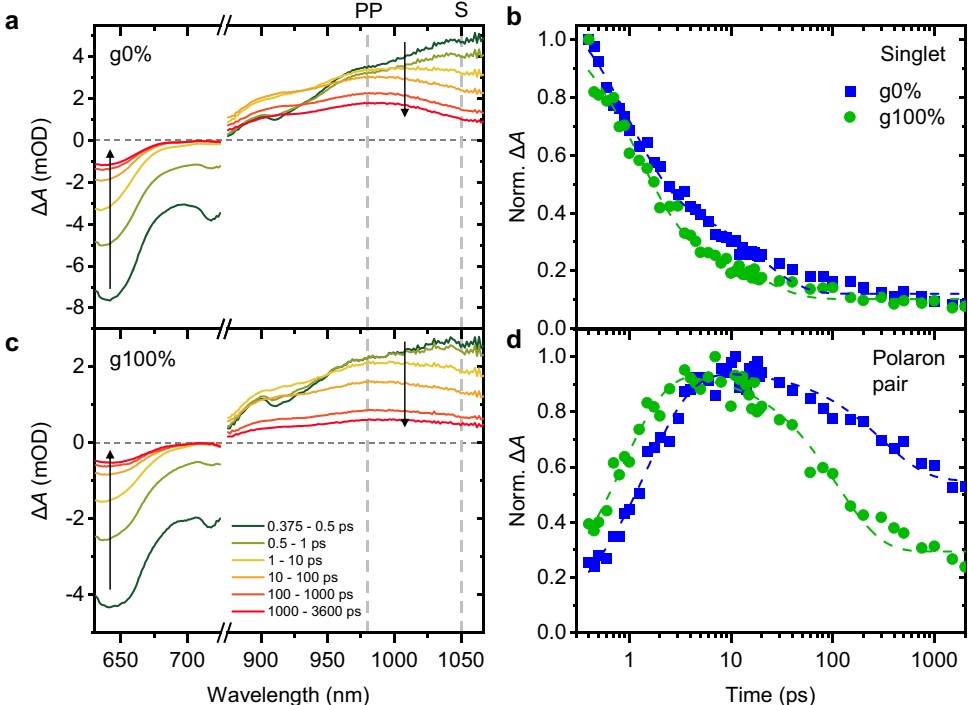

**Fig. 4 | Raw TA spectra of the g0% and g100% polymers.** Shown are raw TA spectra for thin-film samples of the (**a**) g0% and (**c**) g100% polymers at different pump-probe delay times, probed from 630 nm–1060 nm after photoexcitation at 600 nm, 17 µJ cm⁻². Spectra are averaged over the delay times indicated. The negative band corresponds to ground-state bleach (640 nm) and stimulated emission from the excited state (720 nm). The broad, positive band in the NIR region is attributed to the photoinduced absorption of the photoexcited state. The shorter-lived signal centred at ≈1050 nm is attributed to singlet excitons (labelled 'S') and the longer-lived signal centred at ≈980 nm to polaron pairs (labelled 'PP'). Vertical arrows show the direction of evolution of the spectra over time. **b** Normalised kinetics of the deconvoluted singlet TA spectrum, fitted with a biexponential decay (dashed lines). **d** Normalised kinetics of the deconvoluted PP TA spectrum, fitted with a multi-exponential (dashed lines). Source data are provided as a Source Data file.

ultrafast evolution of peak positions and frequencies in the C = C stretching region relate directly to the nuclear relaxation dynamics associated with ES formation in the polymers. Furthermore, FSRS provides a direct means to investigate the relationship between molecular structure, vibrational cooling, and the impact of vibrational coupling on decay processes in conjugated polymers. Unravelling the relationship between electronic and vibrational energy within molecular systems is critical to gain insights into how differences in inner and outer $\lambda$, induced by the glycol sidechains, impact structural changes in the polymer upon photoexcitation, which in turn greatly impact the polymers' device performances.

## Excited-state nature and lifetimes

TAS measurements of thin films of the g0% and g100% polymers were carried out to determine the nature of the photogenerated ES and whether this varies with glycol sidechain content. Transient spectral dynamics were initiated with a 600 nm actinic pump (photoexcitation), which is resonant with the $S_0$-$S_1$ transition of all CPDT polymers (Fig. 1b) and probed over the wavelength range 630-1060 nm. Figure 4 shows the transient absorption (TA) spectra obtained for the g0% and g100% polymers, illustrating the spectral evolution of the polymers up to 3.6 ns. Note that time delays shorter than 0.35 ps were not analysed due to interference effects stemming from the pump-probe temporal overlap.

There are two main features to consider when analysing the TA spectra for both polymers. First, the negative feature at ≈640 nm corresponds to the ground-state bleach (GSB) of the polymers as photoexcitation occurs. The feature at ≈720 nm corresponds to stimulated emission (SE) from the transient ES. Second, there is a broad positive band in the near-IR (NIR) region, which corresponds to

photoinduced absorption (PIA) of the transient ES. Within this broad PIA band, kinetic analysis identifies two distinct spectral dynamics: a short-lived signal centred at ≈1050 nm appearing at early time delays, labelled 'S' in Fig. 4a and c, and a longer-lived signal centred at ≈980 nm, labelled 'PP'. The appearance of the signal centred at 980 nm correlates with the decay of the signal at 1050 nm. Other time-resolved studies on the photoexcitation of thiophene-based polymers show similar spectral features and have been used to aid identification of the ESs forming here[37,46–49]. The short-lived feature at 1050 nm can be assigned to the formation and rapid decay of singlet excitons. The appearance of the longer-lived signal centred at 980 nm is assigned to the formation of polaron pairs, confirmed by evidence of charges at the same spectral position in the microsecond TAS experiments shown in Suppl. Fig. 35. PPs are defined as excitons with increased charge transfer character, with the electron-hole pair being spatially separated and bound with a weaker Coulomb attraction[50–53]. Justification for the assignment of this signal to PPs is outlined below. As this is a thin film homopolymer system, the PP state is likely predominantly intermolecular in nature, being delocalised between packed polymer chains[54]. Given the disordered packing nature of both polymers in the thin film, it is likely that the e·h pair of the PP will be situated on polymer chains in different phases, for example amorphous and crystallised regions. This stabilises the PP state relative to the singlet exciton, similar to how a CT-state in a donor-acceptor blend system is stabilised relative to the singlet state[1]. The microsecond TAS measurements indicate that there are also small amounts of triplet formation (identified using a triplet sensitizer, Suppl. Fig. 35) in the polymers on timescales > 10 ns, longer than those studied in this work. This assignment of the spectral dynamics indicates that both g0% and g100% undergo photoexcitation to the lowest-lying singlet state,

followed by ultrafast PP formation and confirms that the nature of the ES does not change with increasing glycol content on ultrafast timescales.

To further understand the ES dynamics, we apply global analysis (GA) to deconvolute the kinetics of each ES species in the TA spectra. Figure 4b,d shows the kinetics of the deconvoluted TA spectra for the g0% and g100% polymers. The deconvoluted spectra obtained using GA are shown in Suppl. Fig. 36. The results confirm that for both polymers two ES species form upon photoexcitation, singlet excitons and PPs. Singlet kinetics are fit with a biexponential decay and PP signals are fit with a multiexponential fit to guide the eye. The singlet signals show two regions of decay: a fast decay, which occurs in parallel with the growth of the PP signal, and a slower decay at later time (>10 ps). The PP signals grow in with the singlet decay and exhibit longer lifetimes. For both g0% and g100% polymers, singlet decay and PP growth occur on picosecond timescales, which is indicative of ultrafast PP formation occurring following photoexcitation. The assignment of the second ES to PPs is done by considering the fluence dependence of the transient kinetics (Suppl. Fig. 37). The recombination kinetics of the signals show fluence-independent behaviour, which indicates that geminate (monomolecular) recombination dominates in both polymers[55]. We can therefore conclude that bound PPs are forming, not separated polarons (charges). This is further supported by the fact that the charges observed in the microsecond TAS data decay extremely quickly, leaving very small spectral signals with no signatures of long-lived separated polarons (Suppl. Fig. 35e).

Comparing the kinetics of the g0% and g100% polymers in Fig. 4b, d, we see that singlet states decay faster for the g100% polymer. The half-life, $\tau_{1/2}$, of singlet decay is 2.4 ps and 1.7 ps for the g0% and g100% polymers, respectively. In accordance with this, the PP signal of the g0% and g100% spectra reach their maxima in 10 ps and 5 ps, respectively. We can therefore conclude that incorporating glycol sidechains into the backbone accelerates PP formation. Normalising the kinetics of the deconvoluted GA spectra to the maximum of the singlet signal provides some insight into the relative population of PPs that form in each polymer. Suppl. Fig. 35b, d shows the singlet-normalised kinetics for the g0% and g100% polymers. The relative amplitude of the PP signal reaches ≈0.5 and ≈0.6 for the g0% and g100% polymers, respectively, which suggests that a slightly larger PP population forms in the g100% polymer. Enhanced PP formation in the higher glycol content films is consistent with the thin-film photoluminescence (PL) spectra (Suppl. Fig. 38), which show that the PL intensity is quenched with increasing glycol sidechain content.

Enhanced charge formation has been reported previously for conjugated polymers functionalised with glycol sidechains and is typically attributed to an enhanced dielectric constant that reduces exciton binding energy and hence aids exciton dissociation to free charges[56,57]. It is important to note, however, that this glycol sidechain-induced dielectric enhancement typically occurs in the low frequency range and is dominated by nuclear relaxations of the sidechains[21,58]. While this has been shown to be sufficient to suppress non-geminate recombination of charge separated states[59], nuclear relaxations are typically slower than exciton dissociation timescales and mobility of charge carriers[21], which would suggest that dielectric constant enhancement due to the glycol sidechains may have minimal effect on exciton lifetime and charge formation. To confirm this for the CPDT polymers studied here, ellipsometry measurements were used to determine the high-frequency dielectric constants of the polymers. The refractive indices and extinction coefficients are shown in Suppl. Fig. 39 and indicate that there is no measurable enhancement of the high-frequency range dielectric constant with increasing glycol sidechain content. We can therefore exclude the effect of dielectric constant on PP formation. This then raises the question of what is causing the accelerated PP formation rate in the g100% polymer, which is discussed below. Furthermore, for the g100% polymer, the PP signal is

also quenched faster at later times, which implies accelerated geminate recombination rates. This is consistent with its smaller separated charge population on μs timescales compared to g0% (Suppl. Fig. 35e). This faster geminate recombination may be due to the more disordered morphology of the g100% polymer. This will likely inhibit delocalisation of the PPs and hence facilitate faster recombination[60,61]. The larger $\lambda$ of the g100% polymer will also inhibit delocalisation of the PPs as the barrier to charge hopping is larger.

## Ultrafast lattice relaxation dynamics

FSRS spectra of the g0% and g100% polymers in thin films were collected and are shown in Fig. 5a, b. As with the TAS measurements, samples were excited at 600 nm, but an additional 900 nm Raman pump was applied. A Raman pump of 900 nm was chosen as this wavelength is resonant with the singlet excitons and PPs, as is evident from the TAS data (Fig. 4). Raman gain spectra were obtained as described in Methods.

Analogous to TAS difference spectra, FSRS Raman gain spectra have negative and positive peaks. The negative peaks, at 1390 and 1550 cm$^{-1}$, correspond to a decrease in GS population following photoexcitation and match well with the GS Raman spectra in Fig. 3b. These peaks recover over time as the ground state is repopulated. Based on the ground-state Raman assignments, the 1390 cm$^{-1}$ GSB mode is assigned to C = C vibrations along the entire conjugated backbone coupled to the C-C intra-ring mode on the central cyclopentane ring. The 1550 cm$^{-1}$ GSB mode is assigned to outer C = C vibrations on the thiophene rings and the neighbouring inter-unit C-C vibrations. Note that there is a slight shift in the bleaching band positions relative to the GS Raman spectra (Suppl. Fig. 40), which is due to slight imperfect FSRS calibration. As we are interested in tracking changes in the FSRS Raman gain spectra, the impact of this slight difference in the spectra is negligible. There is one positive peak, at ≈1350 cm$^{-1}$, which corresponds to the formation and evolution of the ESs (singlets and PPs) and the structural changes associated with their formation. The position of this ES peak matches expectations from DFT simulations of the $S_1$ and polaronic Raman-gain spectra (Suppl. Fig. 33) and experimental data indicating that there is formation of a Raman peak just below the main neutral Raman mode peak upon ES formation[36,62]. Note that due to the similarities of the $S_1$ and PP Raman spectra (Suppl. Fig. 33), we cannot separate the contribution of the two species to the FSRS ES peak. For this, we use the TAS data, which is described in detail below. We conclude that upon ES formation in the CPDT polymers there are structural changes from benzoidal to quinoidal, permitting π-electron density delocalisation along the backbone, which is associated with backbone planarization. Overall, the C = C bonds lose π-electron density, becoming more single bond in nature, while C-C bonds gain π-electron density, becoming more double bond in nature, resulting in the ES Raman peak shifting to a lower wavenumber[36].

We track two features in the FSRS Raman gain spectra: (1) Relative peak intensities, which, as with steady-state Raman, give us information on π-electron distribution along the conjugated backbone and (2) peak positions, which tell us about structural changes occurring in the backbone and vibrational cooling on the PESs. These help us to understand how the molecular structure is changing over time as ESs form and decay. We first track the relative Raman gain intensities of the GSB and ES peaks and compare these between the g0% and g100% polymers (Fig. 5c–e). For the GSB mode at 1390 cm$^{-1}$, there is no difference between the polymers in the timescales for the peak recovery, $(0.3 \pm 0.04)$ ps for both polymers from biexponential fittings, suggesting that the timescales for structural changes associated with recovery of the GS do not vary with glycol sidechain content. The interesting differences between the polymers become apparent when analysing the ES peak at 1350 cm$^{-1}$. The time taken for the ES peak to grow in increases with increasing glycol sidechain content, with the ES

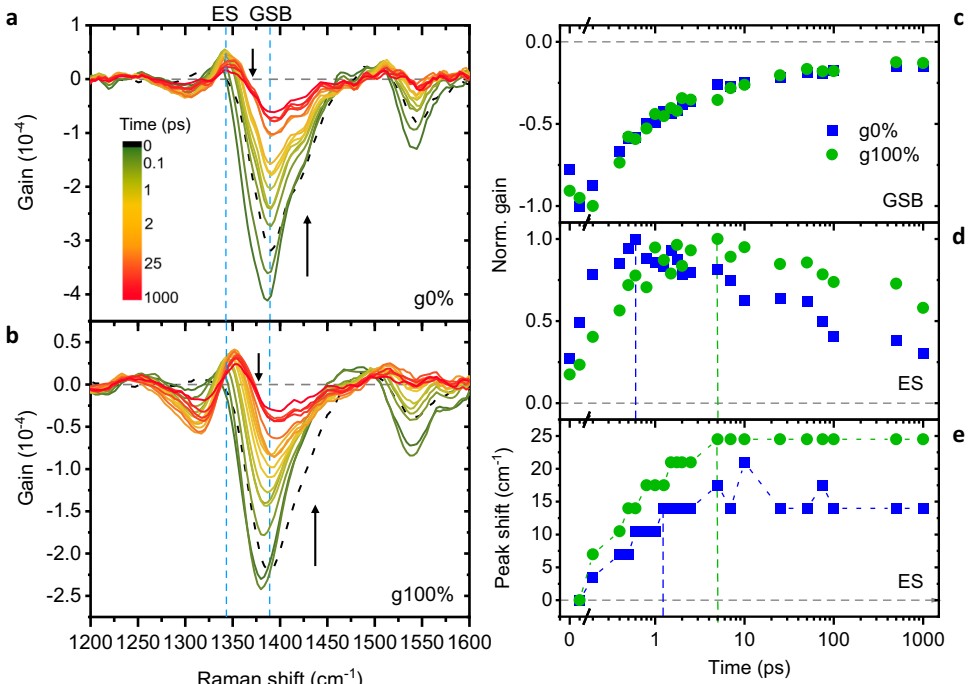

**Fig. 5 | FSRS spectra and extracted kinetics.** FSRS Raman gain spectra of (**a**) g0% and (**b**) g100% polymers in thin films. Negative peaks at ≈1390 and ≈1550 cm⁻¹ are indicative of ground-state bleaching of the polymer upon photoexcitation. The positive peak at ≈1350 cm⁻¹ is indicative of ES formation. The two main modes of interest, 1390 and 1350 cm⁻¹ are labelled GSB (ground-state bleach) and ES (excited state), respectively. Vertical arrows show the direction of evolution of the spectra over time. **c** Time-resolved normalised Raman-gain intensity of the 1390 cm⁻¹ GSB, (**d**) 1350 cm⁻¹ ES modes and (**e**) time-resolved peak shifts of the 1350 cm⁻¹ ES mode. Vertical dashed lines indicate when the maximum relative Raman gain intensities are reached and when ES peak shifting stops for each polymer. Source data are provided as a Source Data file.

peak reaching its maximum relative intensity at (0.6 ± 0.07) ps and (5 ± 0.07) ps for the g0% and g100% polymers, respectively. Additionally, the ES peak is longer-lived in the g100% polymer. The structural relaxation associated with ES formation and decay is, therefore, slower in the higher glycol-content polymer. This is consistent with the higher $\lambda$ measured previously, implying that incorporation of glycol sidechains introduces an energetic barrier to ES formation, therefore increasing the time taken for their formation. This is expanded on below.

Analysis of the ES peak positions supports the above conclusions. It is important to note that our FSRS signal-analysis and baselining methods, described in Methods, ensure that the peak shifts we extract from the FSRS spectra are not artefacts caused by overlapping of the GSB and ES peaks. The extracted peak shifts for the ES peak are shown in Fig. 5e. Peak shifts of the Raman modes to higher wavenumbers at early times occur because of vibrational cooling on the PESs. Electronic excitation populates the F-C state ($S_1$ $\nu_n$ state, where $\nu_n$ is a given vibrational energy level and $n > 1$) and there is immediate relaxation from this nascent F-C state. The polymer cascades down the vibrational energy levels and during the relaxation process, opportunities for structural changes are created that ultimately leads to the minimum of the $S_1$ PES. This thermalisation typically occurs on timescales of picoseconds to tens of picoseconds[63], and is seen as a shift to higher wavenumbers in the FSRS spectra due to the increasing energy gap between the virtual energy state and vibrational energy level on the ES PES as IVR occurs. A more detailed explanation of this is presented in Suppl. Fig. 41. For the ES peak at 1350 cm⁻¹, there is a continuous upshift of the peak for both polymers until plateauing at (1.25 ± 0.07) ps for the g0% polymer and (5 ± 0.07) ps for the g100% polymer. There is an upshift of 14.0 cm⁻¹ and 24.5 cm⁻¹ for the g0% and g100% polymers, respectively. The larger upshift of the peak for the g100% polymer is indicative of more vibrational cooling occurring along the PES of the

g100% polymer, and hence larger degrees of structural change upon relaxation towards the PES minimum, over a given timeframe. The g100% polymer ES peak shift also takes longer to plateau as vibrational cooling, and therefore structural changes, are occurring for longer. This result agrees well with the higher $\lambda$ measured in the g100% polymer, as more vibrational cooling is expected to occur when ground and ES PES minima are more greatly offset, see Fig. 2a.

There is a correlation between the timescales of the FSRS ES peak shifts and the rise in FSRS Raman gain intensities (Fig. 5d, e). When the maximum relative Raman gain intensity of the ES is reached, the peak shift plateaus, i.e., vibrational cooling ends and the minimum of the PES has been reached. For the g0% and g100% polymers, the ES peak maximum is reached at (0.6 ± 0.07) ps and (5 ± 0.07) ps, respectively, and the ES peak shift plateaus at (1.25 ± 0.07) ps and (5 ± 0.07) ps, respectively. Considering this correlation in conjunction with the TAS data provides insight into the ES generation and the subsequent structural relaxation occurring in the polymers. From the TAS data (Fig. 4b, d), at 0.6 ps the singlet population dominates in the g0% polymer. So, the initial structural changes observed in the g0% FSRS spectra prior to ≈1 ps are predominantly associated with the polymer relaxing to the minimum of the $S_1$ PES. However, after the g0% FSRS ES peak maximum is reached at 0.6 ps, the ES peak continues to upshift a further 3.5 cm⁻¹ before plateauing at 1.25 ps, during which time the PP population is growing. The slight further upshift of the ES peak is therefore evidence of small amounts of vibrational cooling to the minimum of the PP state PES occurring. For the g100% polymer, at 5 ps the TAS PP signal has reached its maximum intensity. The structural changes observed in the g100% FSRS spectra, therefore, are associated with both singlet and PP formation, with the formation of PPs and vibrational relaxation occurring on similar timescales.

In summary, in the g100% FSRS spectra we first observe a slower grow-in of the ES peak and more vibrational cooling over a longer

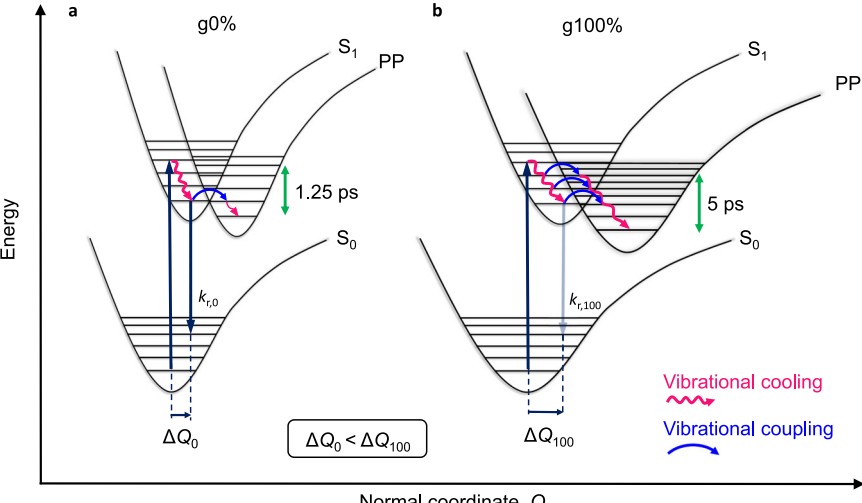

**Fig. 6 | Vibrational coupling-induced PP formation.** PES schematic showing the processes of vibrational cooling (thermalisation) and polaron pair state formation in thin film of (**a**) the g0% polymer, which thermalises in 0.6 ps to the minimum of the $S_1$ state and subsequently forms polaron pairs (labelled PP), thermalising to the minimum of the PP state by 1.25 ps and (**b**) the g100% polymer, where vibrational coupling facilitates PP formation and the polymer thermalises in 5 ps to the minimum of the PP state. Green arrows indicate measured vibrational cooling times. ΔQ is the offset of the $S_0$ and $S_1$ PESs and is larger for the g100% polymer, as shown with the solvatochromic measurements. $k_{r,0}$ and $k_{r,100}$ are the radiative recombination rates for the g0% and g100% polymers, respectively. The PESs of the g100% polymer are shown to be broader than those of the g0% polymer, which is a result of the higher structural disorder in the polymer[39,70]. Source data are provided as a Source Data file.

timeframe. This is consistent with a larger PES offset, greater ES structural relaxation and the higher λ values measured for the g100% polymer. Secondly, the larger degree of vibrational cooling in g100% is consistent with the faster PP formation seen in the deconvoluted TA spectra kinetics of the glycolated polymer. The timescale for the FSRS ES peak growth and vibrational cooling aligns with the growth of the PP signal in the TAS spectra (Suppl. Fig. 42), suggesting that PPs form during structural relaxation of the glycolated polymer. We therefore suggest that the larger degrees of vibrational cooling occurring in the g100% polymer induces stronger vibrational coupling between the singlet and PP states and hence assists ultrafast PP formation.

### Vibrational coupling-induced PP formation

Figure 6 shows PES diagrams of the ES formation and the vibrational coupling in the g0% and g100% polymers. In both polymers, there is photoexcitation to a vibrational energy level (v>0) of the $S_1$ state. The polymer then relaxes out of the Franck-Condon region, thermalising as it undergoes structural changes (planarization and π-electron delocalisation). For the g0% polymer, the structural relaxation to the minimum of the $S_1$ state occurs in 0.6 ps. This is followed by PP formation, after which there is vibrational cooling until 1.25 ps (blue and green vertical dashed lines in Fig. 5e). The PP state reaches its maximum population by 10 ps (Fig. 4d). For the g100% polymer, the slower and larger degree of vibrational cooling means that vibrationally-ESs are populated for longer. This increases the probability of vibrational coupling to the PP state from a hot excitonic state. Therefore, as the polymer thermalises following photoexcitation there is enhanced vibrational coupling between the $S_1$ and PP PESs, which facilitates PP formation. These hot PPs then thermalise by 5 ps, which is the same timescale it takes for PP population to reach its maximum.

Coupling of the TAS and FSRS data has allowed us to track electron-phonon coupling in the CPDT polymers and discern the impact of vibrational coupling, and hence λ, on PP formation. Our results provide direct evidence for the acceleration of PP formation via strong vibrational coupling induced by a larger λ, which is induced by incorporation of glycol sidechains into the conjugated backbone without any modification to the backbone itself.

### Discussion

In this work, we have investigated the effects of incorporating glycol sidechains into the conjugated polymer backbone on the optoelectronic properties of a CPDT polymer series. Glycolation is found to increase the total reorganisation energy of the polymer upon ES formation. Computational calculations and optical solvatochromic measurements show that glycolating the polymer induces a larger inner reorganisation energy upon ES formation. Additionally, steady-state analysis of the polymers reveals that in thin films, increasing glycol sidechain content results in more disordered packing, a reduction in π-π conjugation and hence a more localised π-electron density to within the CPDT unit, which will contribute to a larger outer reorganisation energy. Time-resolved spectroscopic techniques, TAS and FSRS, reveal that the nature of the ES does not vary with glycol sidechain content. However, polaron pair formation is accelerated in the g100% polymer compared to the g0% polymer due to enhanced vibrational coupling between the singlet exciton and polaron pair states. This is attributed to the g100% polymer's higher reorganisation energy upon ES formation resulting in slower and larger degrees of vibrational cooling, which causes vibrational ESs to be populated for longer, hence increasing the probability of vibrationally-assisted polaron pair formation from a hot excitonic state. We hence demonstrate that enhanced polaron pair formation is not due to an enhanced dielectric constant induced by the glycol sidechains, as has often been reported, but is due to enhanced vibrational coupling. Finally, we demonstrate how the FSRS technique used in this work allows us to couple electronic dynamics with the ultrafast lattice structural changes involved in polaron pair formation.

Our results provide a design rule that allows for the tuning of the reorganisation energy via variation of sidechain nature, without modification of the conjugated backbone structure. While a larger reorganisation energy is considered detrimental for applications such as OFETs, where any energetic barrier to charge transport should be minimised, here we demonstrate how such modulation of the reorganisation energy can be used as a tool to control the timescale of photophysical processes such as PP formation. This can be beneficial for applications that require ultrafast and a high density of charge state formation, without optimised charge transport properties. While both minimised reorganisation energy and enhanced PP formation are vital

for device performance, a balance between the two can be struck by using specific percentages of glycol sidechains. The desired application therefore dictates the implementation of the design rule. The ability to correlate polymer structure and reorganisation energy with the nature of photoinduced ES formation provides fundamental understanding for the further development of more efficient optoelectronic devices, in particular for applications where polaron pair (or CT state) formation is critical for determining device operation. This in turn will lead to further improvements in device efficiencies, for example in OPVs where efficient CT-state generation and separation is desired, or hydrogen-evolution photocatalysts where fast formation and high yields of long-lived photogenerated charges are desired[64].

## Methods

### Materials and reagents
All reagents and solvents were purchased from commercial suppliers including Aldrich Inc., Tokyo Chemical Industry Co., Ltd. (TCI), and Alfa Aesar. Anhydrous solvents were dried over molecular sieves 4 Å and degassed prior to use. In particular, 3,3'-dibromo-2,2'-bithiophene (Aldrich Inc, 97%), n-BuLi (Aldrich Inc, 2.5 M in hexanes), dimethylcarbamoyl chloride (Aldrich Inc, 98%), and 13-bromo-2,5,8,11-tetraoxatridecane (TCI, 95%) were purchased and used without purification.

### Polymer characterisation
$^1$H NMR and $^{13}$C NMR spectra were recorded with Advance 300 and DRX 500 MHz FT-NMR Bruker spectrometers. HR mass analysis was obtained by a Joel JMS-700. Molecular weight analyses (calibrated against polystyrene (PS) standards) of the polymers were carried out on an analytical gel permeation chromatography (GPC) Agilent Technologies 1200 series GPC fitted with a 10 μm 50 mm x 7.5 mm guard column, two PLgel 10 μm mixed-B 7.5 mm x 300 mm columns, and refractive index (RI) and UV (254 nm) detectors, running in HPLC-grade chloroform solvent at 40 °C with a flow rate of 1.0 mL min$^{-1}$ at a sample concentration of approximately 1 mg mL$^{-1}$. Thermogravimetric analysis (TGA) was performed on a TA 2050 thermogravimetric analyser under a nitrogen atmosphere at a rate of 10 °C min$^{-1}$. Differential scanning calorimetry (DSC) plots were recorded under nitrogen with a TA Instruments 2100 differential scanning calorimeter. Samples were heated at 10 °C min$^{-1}$ from 0 – 300 °C. Solution and film UV-vis-NIR absorption spectra were recorded by using a Cary 5000 UV-vis-near IR double-beam spectrophotometer. Cyclic voltammetry (CV) was performed at a room temperature in a 0.1 M solution of tetrabutylammonium perchlorate in acetonitrile under nitrogen gas at a scan rate of 50 mV s$^{-1}$. A glassy carbon electrode as the working electrode, a Pt wire was used as the counter electrode and an Ag/AgCl electrode as the reference electrode.

### Sample preparation
The CPDT polymers were dissolved in chloroform at a concentration of 5 mg ml$^{-1}$ and stirred at 40 °C for several hours. Solutions were filtered at 0.45 μm and left overnight, then being spin coated onto cleaned substrates at 2000 rpm for 40 s. Deposited films were left in a desiccator overnight and stored in a nitrogen glovebox. Film thicknesses were measured with a Bruker DektakXT, averaging ≈30 nm for the polymer films. OFET devices for Raman measurements were fabricated using Fraunhofer silicon wafers with a silicon dioxide layer and the source and drain electrodes ($W$ = 2 mm $L$ = 5 μm) formed of a 10 nm ITO adhesion layer and topped with 30 nm of gold. No contacts to the OFET devices were made during Raman spectra acquisition, these samples were used as the gold electrodes enhance the non-resonant Raman spectra intensities. Quartz substrates were used for optical measurements. In addition to the neat films, 1:1 PS blend and 1:1 zinc tetraphenylporphyrin (ZnTPP) blend films were made to study the TA spectra in more detail, with ZnTPP enabling sensitisation experiments where triplet generation is promoted.

### DFT calculations
Geometry analysis, reorganisation energy calculations and Raman peak assignments were performed using DFT and TD-DFT on the Imperial College High-Performance Computing service using GAUSSIAN09 software[65]. Simulations were performed on isolated molecules in the gas phase using B3LYP level of theory and 6-31 G(d,p) basis set[66]. Full geometry optimisation was done for oligomers in the neutral, hole polaron and singlet exciton states. To ensure that the global minimum geometry was reached, a tight convergence criterion was selected in the geometry optimisation calculations. For reorganisation energy calculations, oligomer lengths of 1, 3, and 5 CPDT units with butyl sidechain lengths were simulated to reduce computational time. Glycol sidechain lengths were matched to alkyl sidechain lengths. Reorganisation energy for a given oligomer was calculated using Eq. (1) where $\lambda_1 = E^1g_O - E^1g_1$ and $\lambda_2 = E^0g_1 - E^0g_O$. $E^1g_1$ and $E^0g_O$ are the energies of the optimised $S_1$ and $S_0$ (ground state) oligomers, respectively. $E^1g_O$ and $E^0g_1$ are the energies of the $S_1$ state oligomer in the optimised ground state geometry and the ground state oligomer in the optimised $S_1$ state geometry, respectively, and are calculated by freezing the molecule in the relevant geometry and performing an energy calculation on the oligomer. Raman spectra of CPDT trimers were simulated with an empirical scaling factor of 0.97 and peak assignments were visualised using GaussView 6.0.16 software[67].

### Grazing-incidence wide-angle X-ray scattering
GIWAXS was performed at PLS-II 3C SAXS beamline, Pohang, South Korea. The X-ray photon energy was 9.8 keV ($\lambda$ = 1.27 Å). The sample to detector distance (SDD) was 220 mm. The 2D data was collected by Eiger X4M (Dectris, Switzerland) having 2070 x 2167 pixels with 75 μm$^2$ pixel size. The incidence angle of the X-ray beam was set at 0.16°, which is below the critical angle of silicon substrate.

### UV-vis and fluorescence spectroscopy
The transmittance of thin films on quartz was measured using a Shimadzu UV-2600 13 UV/visible spectrophotometer with integrating sphere attachment. Air was used as a baseline reference and the thin film samples were corrected to the transmission of a clean blank quartz substrate. The absorbance of the films was then obtained using: Absorbance = log($T_{substrate}/T_{sample}$). Solvatochromic measurements were performed using a Shimadzu UV-2550 with solution samples at a concentration of ≈0.01 mg mL$^{-1}$. Transmittance spectra were taken for each polymer in toluene, chlorobenzene, chloroform and dichloromethane. Air was used as a baseline reference and the solution samples were corrected to the transmission of the corresponding solvent. The absorbance of the solution sample was then obtained using the same equation as previously stated. Fluorescence spectra of the solution samples were measured with a Shimadzu RF-5301PC fluorescence spectrometer at room temperature.

### Steady-state Raman spectroscopy
PL and Raman spectra were measured with a Renishaw inVia Raman microscope in a backscattering configuration with an InGaAs detector and a 50 × objective. A diffraction grating of 1200 lines mm$^{-1}$ was used for Raman measurements and a 300 lines mm$^{-1}$ grating was used for PL measurements. During measurement, samples were placed in a Linkam stage and were under constant nitrogen flow. Raman spectra were collected using a 785 nm diode laser at 9.0 mW power and the laser probe was defocused to ≈10 μm. PL background was removed from the Raman spectra with polynomial fitting. PL spectra were collected using a 514 nm argon-ion laser with the laser probe defocused to ≈10 μm. Laser powers and acquisition times were optimised to give the best spectra but were kept consistent between samples. Raman spectral calibration was performed using the well-defined 520 cm$^{-1}$ peaks of a Si reference.

## µs transient absorption spectroscopy

A pump-probe micro-to-millisecond TA spectroscopy set-up was used to measure the TA spectra and kinetics. Laser pulses (repetition rate 10 Hz, pulse duration 6 ns) were generated by a Nd:YAG laser (Spectra Physics, INDI-40-10). Excitation wavelengths (600 nm for neat and PS blend films, 430 nm for ZnTPP blend films) were selected by a ver-saScan L-532 OPO. The ZnTPP blend films were excited at the ZnTPP Soret band to promote the generation of triplets. The excitation density was set in the range between 3 and 140 µJ cm$^{-2}$ using neutral density filters, measured by a ES111C Thorlabs power meter (with TA spectra measured at 12 µJ cm$^{-2}$). The probe light was provided by a quartz tungsten halogen lamp (IL1, Bentham). Probe wavelength selectivity was achieved using bandpass filters and a Cornerstone 130 mono-chromator (Oriel Instrument) before the detector. The TA signals were recorded with Si and InGaAs photodiodes. The signal from the pho-todiodes was preamplified and sent to the main amplification system with an electronic filter (Costronic Electronics), which was connected to an oscilloscope (Tektronics, DPO4034 B) and PC.

## Ultrafast transient absorption spectroscopy and fs stimulated Raman spectroscopy

FSRS was carried out on the ULTRA set-up[68] at the Central Laser Facility at the Rutherford-Appleton Laboratory. FSRS uses three pulses, an actinic pump (600 nm, ≈50 fs), a Raman pump (900 nm, 1.5 ps) and a broadband probe pulse (≈50 fs), which yields an instrument response of ≈100 fs. These were generated using two titanium-sapphire ampli-fiers (Thales Laser) seeded from a single oscillator (20 fs, Vitara-T-HP from Coherent) to create synchronised femtosecond and picosecond outputs after compression, respectively. The femtosecond beam was split into two, with one beam pumping an optical parametric amplifier (Light Conversion TOPAS) to generate the actinic pulse, and the other beam being focused onto a sapphire window to generate a white light continuum used for the broadband probe pulse. Notch filters (Kaiser Holographic notch plus) were used to reject the 800 nm fundamental wavelength of the laser from the white light continuum. A small frac-tion of the probe beam was split before the sample to serve as a reference beam. The Raman pump pulse was generated by tuning the picosecond beam to 900 nm with an optical parametric amplifier (Light Conversion TOPAS). The probe beam was kept at the system repetition rate of 10 kHz, while the actinic and Raman beams were mechanically chopped at 5 and 2.5 kHz, respectively. The beams' dif-ferent repetition rates allowed for four pulse sequences: actinic-probe (01), Raman-probe (10), actinic-Raman-probe (11) and probe (00). Transient absorption spectra were obtained with the actinic-probe and probe sequences and FSRS spectra were obtained with a combination of the four sequences[68]. A linear motor drive translation stage (New-port) provided femtosecond to nanosecond pump-probe timing. Beams were focused onto samples with beam diameters of 50 µm (probe beam) and 100 µm (actinic and Raman beams). The beams were set in a non-collinear geometry, with 10° between the Raman and actinic beams and the probe beam. The CPDT polymers were spin coated onto calcium fluoride windows, which were placed into a Har-rick cell (Harrick Scientific) and were under constant nitrogen flow during measurements. The probe and reference beams were colli-mated and focused into home-built diffraction spectrographs (0.25 m, $f$/4) and detected using a high-rate readout linear detector (Quantum Detectors). FSRS spectra were baseline-corrected using a polynomial fit to the background signal, leaving the FSRS peaks unchanged[69]. To ensure that the extracted peak dynamics were not caused by overlap of the ground and excited state peaks, the corresponding ground state Raman spectrum of the polymer was subtracted from its baselined FSRS spectra. Peak positions were extracted from the resulting spec-tra. The ground state Raman spectra subtracted from the FSRS spectra were obtained during FSRS measurements, when only the Raman-probe (10) was illuminating the sample.

## Ellipsometry

The optical constants of the thin polymer films were determined from measurements at several angles of incidence using a rotating polariser GES5E ellipsometer from SOPRALAB with a spectral range from 1.2 – 5.5 eV. All regression analyses were performed using both an in-house code and the Winelli2 software from SOPRALAB. The former was used to consider the possibility of optical anisotropic behaviour in the films, which was found to be negligible in all samples. Therefore, the optical model was built in Winelli2 by parameterising the isotropic films dielectric constants as standard sums of critical points representing the joint density of electronic states. The quartz substrate was mea-sured separately and used as reference in the model.

## Data availability

The data that support the findings of this study are available from the corresponding author upon request. Source data are provided with this paper.

## Code availability

The code used are publicly available via ref. 69. and from the corre-sponding author upon request.

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

## Acknowledgements

The authors are thankful for the opportunity and funding made available by the STFC and UKRI for access to the Central Laser Facility (App. #22130036) and would like to acknowledge all CLF staff who aided us with our measurements during our beamtime. The authors would like to thank Prof. James Durrant for insightful discussions and Dr. Sam Hillman for TAS analysis guidance. K.P., J.L., E.T., K.S., and J.S.K. acknowledge the UK EPSRC for studentships under DTG and the Processable Electronics Centre for Doctoral Training (EP/L016702/1). This research was also supported by the UK Department for Business, Energy and Industrial Strategy (BEIS) through the National Measurement System and the Val O'Donoghue PhD Scholarship at Imperial College London. J.L. and J.S.K acknowledge funding from the UK EPSRC ATIP Programme Grant (EP/T028513/1). Y.H.K., J.K. and N.Y.K. acknowledge National Research Foundation grant NRF (RS-2023-00301974) and NRF (RS-2024-00336766) for financial support. T.M.C. and H.I.G. would like to acknowledge support from the EPSRC (EP/N509577/1, EP/T517793/1). B.D., M.I.A. and M.C.Q. thank the financial support from the Spanish Ministry of Science and Innovation through projects PID2021-128924OB-I00 and CEX2019-000917-S (FUNFUTURE, Spanish Severo Ochoa Centre of Excellence program) and Generalitat de Catalunya (2021-SGR-00444).

## Author contributions

K.P. conceived the study, performed UV-vis, steady-state Raman spectroscopy, ultrafast TAS and FSRS measurements, performed DFT calculations, analysed data and prepared the first draft of the manuscript. J.L. helped conceive the study, assisted with ultrafast TAS and FSRS measurements, data analysis, interpretation of results and assisted with the first draft of the manuscript. J.G.K. and N.Y.K. synthesised the polymer series and performed chemical characterisation under the supervision of Y.H.K. E.T. performed solvatochromic measurements on the polymers and assisted in data analysis and preparation of the manuscript. K.S. assisted in sample preparation for ultrafast TAS and FSRS measurements and preparation of the manuscript. I.V.S, G.K. and A.W.P. assisted with ultrafast TAS and FSRS measurements, provided guidance for data analysis and assisted in interpretation of results and preparation of the manuscript. H.I.G. performed microsecond TAS measurements under the supervision of T.M.C and both assisted in the interpretation of the data and preparation of the manuscript. A.V.M. performed GPC measurements. S.K. and Y.Y.K. performed GIWAXS measurements. M.C.Q., M.I.A. and B.D. performed ellipsometry measurements. J.S.K. conceived the study, procured funding, assisted in the drafting of the manuscript, and supervised the project. All authors contributed to the final version of the manuscript.

## Competing interests

The authors declare no competing interests.
