## [Peer Review File · Nature Communications]

Slow vibrational relaxation drives ultrafast formation of photoexcited polaron pair states in glycolated conjugated polymersREVIEWER COMMENTS

Reviewer #1 (Remarks to the Author):

This manuscript reports the effects of glycol sidechains on the electronic states and electronic properties of a series of cyclopentadithiophene (CPDT) polymers, which have varying proportions of monomer units with glycol sidechains. Increasing glycol sidechain content results in more disordered packing, a reduction in π - π conjugation and hence a more localized π -electron density to within the CPDT unit. Especially, glycolation is found to increase the total reorganisation energy of the polymer upon ES formation, slower and larger degrees of vibrational cooling, and thus enhanced vibrationally-assisted polaron pair formation from a hot excitonic state, which is not due to an enhanced dielectric constant induced by the glycol sidechains, as has often been reported. The results are very interesting and the FSRs technique used in this work is important. Thus, I recommend this manuscript be accepted after revisions as suggested below:

1. The authors studied a series of cyclopentadithiophene (CPDT) polymers, which have varying proportions of monomer units with glycol sidechains (0%, 25%, 50% and 100%). I recommend to add a polymer with 75% proportion of monomer units with glycol sidechains to provide a complete scan of the series of CPDT polymers
2. According to SI, the Mw of CPDT-g(100%) is only 39 kDa, while the Mws of the other polymers are much higher (1022 kDa, 817 kDa and 697 kDa). Since the molecular weight is a very important factor for packing in solid state. I also recommend to synthesize a new CPDT-g(100%) with similar high molecular weight to repeat the investigation in solid film, which may also reveal the effect of molecular weight on the total reorganisation energy and polaron pair formation.
3. In Figure 2b, the DFT-calculated λ values for a monomer, trimer and pentamer of the g0% and g100% polymers are shown. I recommend to add the results of the g50% polymer at least to confirm the effect of the glycol sidechains.
4. In Figure 2c, the experimental Stokes shift values (top) and Huang-Rhys Factors (bottom) of CPDT-g(0%, 50% and 100%). I feel very strange why there is no Stokes shift values (top) and Huang-Rhys Factors of CPDT-g25%, which I recommend to add the results of CPDT-g25% in the figure as well.
5. For Figure 3c, the relative peak intensities of peaks 1 and 3 for the CPDT polymers from g0% to g100%. Are the values of X axis correct? It is the same distance between g25%-g50% and g50%-g100%.

Reviewer #2 (Remarks to the Author):

This paper "Slow vibrational relaxation drives ultrafast formation of photoexcited polaron state in glycolated conjugated polymers" systematically explored the glycol sidechain effect on excited state formation in CPDT polymers from steady-state properties and kinetic characterization, which is necessary for guiding the optoelectronic materials design. Besides, it emphasizes the role of the structure relaxation process in organic polymers from the photophysical perspective allowing us to comprehensively evaluate the weight of related parameters. The work merits publication. However, in the manuscript, there are some questions and data analysis problems that need to be discussed or revised.

1. As the author mentioned in the introduction part, minimizing the reorganization energy to decrease the energy barrier to excited state and charge formation is necessary for organic optoelectronic devices. In this work, the introduction of glycol sidechain leads to higher reorganization energy, more disorder, and localized electron, but can accelerate the polaron pairs formation. How to balance the trade-off between the fast polaron formation and the related side effects of larger reorganization energy in material design? A potential direction can be added in the discussion part.
2. Compared with alkyl sidechains, the glycol sidechains present some advantages in the introduction

part. But, for example in the organic photovoltaic field, most of the donor and acceptor materials are based on alkyl sidechains. The reason for the lagging development of glycol sidechains should be added in the introduction part.

3. In the reorganization energy calculation part, why the Stokes shift and Huang-Rhys factor information of g25% are missing in Figure 2(c)? It should be discussed as g25% indicates more ordered aggregation in Figure 1.

4. How about the polymer molecular weight difference by increasing the glycol sidechain content? It is also important for photophysical properties in solutions and solid films and should be discussed.

5. In the part of steady-state Raman spectroscopy, the author only discussed the difference in peak intensity for polymers. How about the effect of the variation of peak shape?

6. It would be more convincing to elaborate on the localization of n -electron density in the backbone by cooperating with theory calculation or other characterization.

7. From kinetic analysis, how could we distinguish the effect between disorder and polaron formation on faster singlet decay and photoluminescence quenching in g100% polymer thin film?

8. More description and explanations should be added in the supporting information for the second part materials properties (Figure S25-S27, and Table S1).

9. Format and details of supporting information should be revised.

Reviewer #3 (Remarks to the Author):

Reviewer #4 (Remarks to the Author):

The paper by Pagano and coauthors describes the effect of polymer glycolation (when alkyl sidechains are replaced by glycol sidechains) on the optoelectronic properties of a series of cyclopentadithiophene (CPDT) conjugated polymers. They observe an enhanced polaron formation in polymers with larger glycol sidechain content. While such an effect is commonly attributed to an enhanced dielectric constant, the authors here attribute it to higher reorganization energy and enhanced vibrational coupling between singlet excitons and polaron pair states. This is the key result of the paper. If correct, then this is an important conclusion.

The observation of an enhanced polaron formation upon glycolation is fully supported by their experiments. The authors performed an impressive set of measurements in a detailed and careful manner, including ultrafast transient absorption spectroscopy and femtosecond stimulated Raman spectroscopy, to monitor the structural relaxation dynamics of their polymers upon photoexcitation. Their arguments to correlate the enhanced polaron formation with enhanced vibrational coupling are plausible. I am, however, not sure whether I am at present fully convinced by their assignment to reorganization energy, though this is a point that may be addressed in a revised version. My general opinion is that the paper describes a very interesting study which is of considerable interest to the field, hence I recommend publication once the following points have been addressed

1. To support their correlation between increased polaron pair formation and reorganization energy, the authors quote (in Fig. 2c) the Stokes' shift and Huang-Rhys parameter, as well as calculations for oligomers up to length of 5 units. The calculations indeed show larger reorganization energies for the glycolated compounds, yet straight line fits through both sets of data point indicate that the energies converge for oligomers longer than about 7 repeat unit, so this only holds if the conjugation length is sufficiently short. The Stokes' shift is read out from Fig. S30. This is a central argument and therefore this figure should be placed in the main manuscript. It would also help if it was plotted on an energy scale and with a range that actually allows reading off values. At present – on a wavelength scale from 500 to 800 nm – it is not possible to judge the validity of the data presented in Figure 2c (the distances between the 0-0 peaks differ by less than 1 nm in a A4 printout). Similarly, I am doubtful about the Huang-Rhys parameter data. In different solvents, the spectra have different linewidth, and the 0-0 and 0-1 peak overlap. So just taking the ratio between them may give erroneous values. It may be that their data supports the conclusions, yet from the current presentation of the data it is not possible to assess this. It would help if the authors could decompose the spectra at least into a sum of Gaussian peaks with equal width to clearly identify the 0-0 positions of absorption and emission, and the correct height of the 0-0 and 0-1 peak. When doing this, it is important to plot the reduced absorption and emission spectra, to correctly identify the geometric reorganization energy (as half of the distance between the two 0-0 peaks). A full Franck-Condon Analysis would, of course, be even better, in particular since the Raman data for the materials are available.

2. The assignment to vibrational coupling is based on the time resolved PAS and Raman spectra, and this seems convincing. My only point of critique relates to the relative position of S1 and the PP state in Figure 6. The authors present data for a single compound in solution. For a single compound, the charge-transfer state that forms upon transferring an electron to an adjacent chromophore is usually at higher energy than the singlet state where it originates from, due to the coulomb attraction that needs to be overcome when separating out electron and hole (and that is much larger than any reduction due to additional reorganization energy) . This is currently not reflected in figure 6, where the PP minimum is shown below the S1 state. Perhaps this could be corrected.

Minor issues:

3. Line 157: The authors claim that " λ is the energy required for the molecular lattice to distort upon ES formation", however, in fact only λ_1 is required for that. The same comment is applicable to Eq.(1) – only λ_1 energy is needed to form ES. The authors may wish to correct this

4. Lines 490-491: A reference to Fig.5(b), by which the values of 0.6nm and 1.25ns were obtained, would be helpful.

5. In Fig. 4 and 5, longer ticks would be very helpful to read off values (in particular for the log scale in Figure 4b)

6. Line 340: What does GA stand for?

Reviewer #5 (Remarks to the Author):

Responses to the reviewers' comments

We thank all reviewers for their time to read the manuscript and for their encouraging and constructive comments. We believe that we have addressed all of the reviewers' comments. Below are our detailed replies to the reviewers' comments and questions. Our detailed point by point responses are in blue and the changes in the manuscript are highlighted in yellow. We hope that our revised manuscript is now acceptable for publication in *Nature Communications*.

Reviewer #1 (Remarks to the Author):

This manuscript reports the effects of glycol sidechains on the electronic states and electronic properties of a series of cyclopentadithiophene (CPDT) polymers, which have varying proportions of monomer units with glycol sidechains. Increasing glycol sidechain content results in more disordered packing, a reduction in π - π conjugation and hence a more localized π -electron density to within the CPDT unit. Especially, glycolation is found to increase the total reorganisation energy of the polymer upon ES formation, slower and larger degrees of vibrational cooling, and thus enhanced vibrationally-assisted polaron pair formation from a hot excitonic state, which is not due to an enhanced dielectric constant induced by the glycol sidechains, as has often been reported. The results are very interesting and the FSRS technique used in this work is important. Thus, I recommend this manuscript be accepted after revisions as suggested below:

Our response: We thank the reviewer for their encouraging comments stating that 'the results are very interesting and the FSRS technique used in this work is important'.

1. The authors studied a series of cyclopentadithiophene (CPDT) polymers, which have varying proportions of monomer units with glycol sidechains (0%, 25%, 50% and 100%). I recommend to add a polymer with 75% proportion of monomer units with glycol sidechains to provide a complete scan of the series of CPDT polymers.

Our response: A g75% polymer was synthesised as part of the CPDT polymer series and included in the preliminary, steady-state characterisation of the polymers. The thin film UV-Vis and Raman spectra for the full series are shown in Figure R1 and Figure R2 below and in both sets of data the g75% polymer lies out of trend. This is mainly due to its significantly lower molecular weight (Mn 13 kDa and Mw 24 kDa) compared to other polymers (Mn 20-23 kDa and Mw 49-66 kDa, see Table 1). In the UV-Vis absorption, the g75% spectrum maximum shows the largest blueshift relative to the g0% spectrum and a large loss in vibronic features, which suggests that the g75% film has the most disordered packing structure with low levels of crystallinity, likely induced by its low Mw rather than the fraction of glycol side chains. Similarly, the g75% Raman spectrum lies out of trend, with peaks 2 and 3 showing the lowest relative intensities. Because of this, the g75% data was not included in the manuscript as it was believed that there were synthesis issues with the polymer. The molecular weights of the g0%, g25%, g50%, g75% and g100% polymers were remeasured at higher temperature (40°C) in chloroform, as is detailed below in response to Q2. The higher temperature prevents aggregation effects impacting the measurements and hence yields more accurate values of molecular weight. The results show that compared to the g0%, g25%, g50% and g100% polymers, the g75% has a significantly lower molecular weight. The g75% polymer being an outlier in molecular weight explains the g75% data falling out of trend and justifies our exclusion of it from the steady-

state characterisation of the CPDT polymer series. It is outside of the scope of this work to re-synthesise a higher molecular weight g75% polymer and we believe that the trends seen in the data and conclusions drawn from them hold well without the g75% polymer and thus its inclusion is not essential.

Figure R1: Normalised thin-film UV-Vis spectra of the CPDT polymer series, including the g75% polymer, which shows more blueshift relative to the g0% spectrum and loss of vibronic features than the g100% spectrum. The g75% spectrum clearly lies out of trend, maximum shows the largest blueshift relative to the g0% spectrum and has a large loss in vibronic features, which we attribute to its much lower Mw.

Figure R2: Raman spectra of the CPDT polymer series, including g75%, normalised to the 1380 cm^{-1} mode. The g75% clearly lies out of trend, with peaks 2 and 3 having the lowest relative intensities. We attribute this to g75% having a much lower Mw.

2. According to the SI, the Mw of CPDT-g(100%) is only 39 kDa, while the Mws of the other polymers are much higher (1022 kDa, 817 kDa and 697 kDa). Since the molecular weight is a very important factor for packing in solid state. I also recommend to synthesize a new CPDT-g(100%) with similar high molecular weight to repeat the investigation in solid film, which may also reveal the effect of molecular weight on the total reorganisation energy and polaron pair formation.

Our response: We thank the reviewer for pointing out the importance of the molecular weight of the conjugated polymers. The initial GPC measurements were conducted at room temperature in chloroform solvent and yielded large PDIs for many of the polymers. This likely occurred due to aggregation effects because of the low temperature that the measurements were conducted in. To ensure an accurate measure of the molecular weight given its importance to our conclusions, we have enrolled the help of specialists in the GPC of polymers to remeasure the molecular weights. GPC measurements were done at 40°C in chloroform, the elevated temperature improves reliability of the measurement by reducing aggregation effects. The results are summarised in the table below and show that the g0%, g25%, g50% and g100% polymers have very similar molecular weights. We can therefore be confident that the results presented, and conclusions drawn throughout the manuscript do not arise from molecular weight differences.

Table R1: Summary of the Mn and Mw values for the CPDT polymer series, measured with GPC at 40°C in chloroform.

Polymer	Mn (kDa)	Mw (kDa)	PDI
g0%	22	49	2.2
g25%	22	50	2.3
g50%	23	55	2.4
g75%	13	24	1.8
g100%	20	66	3.4

We have updated the polymer molecular weights in the supplementary information, section 1 (Experimental Synthesis), which are highlighted in the text (Lines 148-179), and added a summary table of the measurements to section 1 (Line 251):

Table S1: Summary of the Mn and Mw values for the CPDT polymer series, measured with GPC at 40°C in chloroform.

Polymer	Mn (kDa)	Mw (kDa)	PDI
g0%	22	49	2.2
g25%	22	50	2.3
g50%	23	55	2.4
g100%	20	66	3.4

We have added Adam Marsh as a co-author on the paper as he carried out the GPC measurements. Line 4 and 17:

Adam V. Marsh,⁵

3. In Figure 2b, the DFT-calculated λ values for a monomer, trimer and pentamer of the g0% and g100% polymers are shown. I recommend to add the results of the g50% polymer at least to confirm the effect of the glycol sidechains.

Our response: We acknowledge the reviewer's suggestion and performed further calculations with smaller contents of glycol sidechains. We simulated trimers of g30% and g60% to clarify the reorganisation energy changes upon glycolation in DFT. We use the ground-state dihedral angles as a guide. The average ground-state dihedral angle is generally the largest component determining reorganisation energy values in DFT calculations. Hence, we use the average ground-state dihedral angle to predict the trend in reorganisation energy. Figure R3 shows the average ground-state dihedral angle for the trimers, which shows that the average ground-state dihedral angle increases with increasing glycol content. This is indicative of an increasing singlet reorganisation energy as a larger dihedral angle change is required when the singlet is formed. As the computational simulations are intended as guidelines, we believe that showing the two extremes in the trend in the manuscript is sufficient to support the experimental data that shows clear evidence of increasing reorganisation energy with glycol content. Note that the full calculations of the singlet state and hence the reorganisation energy values in these trimers require a long computational time without providing any additional information.

Figure R3: Average ground-state dihedral angles of g0%, g30%, g60% and g100% trimers simulated in DFT.

4. In Figure 2c, the experimental Stokes shift values (top) and Huang-Rhys Factors (bottom) of CPDT-g(0%, 50% and 100%). I feel very strange why there is no Stokes shift values (top) and Huang-Rhys Factors of CPDT-g25%, which I recommend to add the results of CPDT-g25% in the figure as well.

Our response: We thank the reviewer for their comment and while we agree that it would be good to have the Stokes shift and Huang-Rhys factor values for the g25% polymer for completeness, unfortunately, it was not possible to obtain this data. There was very limited g25% material remaining and using it for the repeat GPC measurements was prioritised. We attempted to measure the PL and absorption spectra with some old solutions, but the results were anomalous due to solution deterioration. Synthesis of an additional batch of polymers with the same Mw is beyond the scope of this work and so although we agree with the reviewer that it would be good to have the g25% data, we do not believe that it would provide further improvement on the conclusions made in the paper.

5. For Figure 3c, the relative peak intensities of peaks 1 and 3 for the CPDT polymers from g0% to g100%. Are the values of X axis correct? It is the same distance between g25%-g50% and g50%-g100%.

Our response: We thank the reviewer for bringing this to our attention and the graph has been corrected by addition of the axes breaks (Line 278):

Figure 3: (a) Raman peak assignments of the two vibrational modes of interest. Peak 1, at around 1370 cm^{-1} , corresponds to the C-C intra mode on the cyclopentane ring. Peak 4, at around 1520 cm^{-1} , corresponds to the C=C intra thiophene mode, and the neighbouring C-C inter mode between the CPDT units. (b) Raman spectra for the CPDT polymers, normalised to the 1380 cm^{-1} mode peak 2*, thin films on prefabricated Fraunhofer OFET substrates, using 785 nm excitation wavelength. (c) Relative peak intensities of peaks 1 and 4 for the CPDT polymers.

Reviewer #2 (Remarks to the Author):

This paper "Slow vibrational relaxation drives ultrafast formation of photoexcited polaron state in glycolated conjugated polymers" systematically explored the glycol sidechain effect on excited state

formation in CPDT polymers from steady-state properties and kinetic characterization, which is necessary for guiding the optoelectronic materials design. Besides, it emphasizes the role of the structure relaxation process in organic polymers from the photophysical perspective allowing us to comprehensively evaluate the weight of related parameters. The work merits publication. However, in the manuscript, there are some questions and data analysis problems that need to be discussed or revised.

Our response: We thank the reviewer for their very positive and encouraging comments.

1. As the author mentioned in the introduction part, minimizing the reorganization energy to decrease the energy barrier to excited state and charge formation is necessary for organic optoelectronic devices. In this work, the introduction of glycol sidechain leads to higher reorganization energy, more disorder, and localized electron, but can accelerate the polaron pairs formation. How to balance the trade-off between the fast polaron formation and the related side effects of larger reorganization energy in material design? A potential direction can be added in the discussion part.

Our response: We thank the reviewer for this very interesting suggestion to provide material design guidelines based on our study. How to balance the trade-off between the fast polaron pair formation and the related side effects of larger reorganization energy is one of the key considerations for conjugated polymers for optoelectronic applications. In this work, we aim to reveal the intramolecular and intermolecular impact of incorporating glycol sidechains into the conjugated backbone. Through simulation and experiment we show that incorporation of glycol sidechains leads to a higher singlet state reorganisation energy, which subsequently allows for accelerated polaron pair formation due to enhanced vibrational coupling to the polaron pair state. Our results present a design rule that allows for tuning of the reorganisation energy and accelerated polaron pair formation without the need for modification of the conjugated backbone structure. As pointed out by the reviewer, this balance between increasing reorganisation energy, which is detrimental to charge transport, and enhanced polaron pair formation does complicate dictating design rules. It is therefore vital to consider the application when defining design rules.

In the alkylated and lower glycol content polymers, the lower degree of polaron pair formation is balanced by the associated lower reorganisation energy, which is beneficial in applications such as organic field effect transistors (OFETs) where the energetic barrier to charge transport should be minimised. In contrast, as our results show, a higher reorganisation energy facilitates accelerated polaron pair formation such as in the g100% polymer. Therefore, glycolated polymers will be more useful when higher or faster charge generation is required, for example in hydrogen-evolution photocatalysts. While a larger reorganisation energy is considered detrimental for certain applications such as OFETs, here we demonstrate that it can be surprisingly beneficial for other applications that require fast and efficient polaron formation. The higher reorganisation energy allows more time for structural relaxation, which allows for efficient polaron pair formation during the relaxation.

This demonstrates that the balance required will depend on the application and that a different percentage content of glycol sidechains will allow for the control of this balance. The incorporation of glycol sidechain is one molecular design rule for the control of reorganisation energy and there are other studies showing that reorganisation energy can be controlled via other molecular design mechanisms. For example, our previous work shows that a lower sidechain density along the

conjugated backbone lowers polaron reorganisation energy and hence facilitates an alternative mechanism of polaron formation.¹

The discussion has been revised to emphasise and clarify this point (Line 555):

Our results provide a design rule that allows for the tuning of the reorganisation energy via variation of sidechain nature, without modification of the conjugated backbone structure. While a larger reorganisation energy is considered detrimental for applications such as OFETs, where any energetic barrier to charge transport should be minimised, here we demonstrate how such modulation of the reorganisation energy can be used as a tool to control the timescale of photophysical processes such as PP formation. This can be beneficial for applications that require ultrafast and a high density of charge state formation, without optimised charge transport properties. While both minimised reorganisation energy and enhanced PP formation are vital for device performance, a balance between the two can be struck by using specific percentages of glycol sidechains. The desired application therefore dictates the implementation of the design rule.

2. Compared with alkyl sidechains, the glycol sidechains present some advantages in the introduction part. But, for example in the organic photovoltaic field, most of the donor and acceptor materials are based on alkyl sidechains. The reason for the lagging development of glycol sidechains should be added in the introduction part.

Our response: We thank the reviewer for their suggestion.

Organic semiconductors intrinsically have low dielectric constants, which limits their efficiency in devices such as organic solar cells. Photogenerated excitons are strongly Coulombically bound and therefore recombine easily, limiting charge generation. There has therefore been interest in incorporating glycol sidechains into donor-acceptor type OPV materials as a mechanism to enhance the material dielectric constant and improve charge generation for photovoltaic applications.^{2,3} However, in photovoltaics charge generation, transport and recombination processes are equally important and all need to be optimised for good device performance. While incorporation of glycol sidechains has been shown to modulate the dielectric constant (mostly at low frequency kHz-MHz ranges), their role for charge generation is unclear. The incorporation of glycol sidechains can often disrupt the packing and order of the polymers, which can worsen charge transport properties and recombination losses, and hence leading to lower device performance. So, the overall performance of glycolated materials has not seen considerable gains in the OPV field and as such their development has not been prioritised.

More importantly, there are synthesis challenges that have inhibited the fast development of glycolated donor-acceptor type OPV materials. Synthesis of the glycol sidechain itself in addition to the synthesis process of incorporating the glycol sidechains into the donor-acceptor type materials require many more steps than with alkyl sidechains, and the synthesis is often not possible. This has resulted in very limited synthesis of donor-acceptor type OPV materials incorporating glycol sidechains.

Text has been added to the introduction summarising these points (Line 77):

The incorporation of glycol sidechains into donor and acceptor materials, or donor-acceptor type non-fullerene acceptors (NFAs) and copolymers typically used in OPVs, however, lags the development of glycolated homopolymers. This is because the benefit of an enhanced dielectric constant, which can improve charge generation, can be accompanied by poorer charge transport

properties. For photovoltaics, charge generation, transport and recombination processes are equally important and all need to be optimised for good device performance. Therefore, as photovoltaic applications have been the driving force for the development of glycolated donor, acceptor and donor-acceptor type materials, the need for good charge transport properties in addition to enhanced charge generation has meant that their development is slow. In addition to this, the complicated synthesis process of incorporating glycol sidechains into the donor and acceptor materials requires many more steps than with alkyl sidechains and is often not possible. This has further limited the development of glycolated OPV materials.

3. In the reorganization energy calculation part, why the Stokes shift and Huang-Rhys factor information of g25% are missing in Figure 2(c)? It should be discussed as g25% indicates more ordered aggregation in Figure 1.

Our response: We thank the reviewer for their comment. As we responded to reviewer 1's comment 4, while we agree that it would be good to have the Stokes shift and Huang-Rhys factor values for the g25% polymer for completeness, it was not possible to obtain this data. There was very limited g25% material remaining and using it for the repeat GPC measurements was prioritised. We attempted to measure the PL and absorption spectra with some old solutions, but the results were anomalous due to solution deterioration. Synthesis of an additional batch of polymers with the same Mw is beyond the scope of this work and so although we agree with the reviewer that it would be good to have the g25% data, we do not believe that it would provide further improvement on the conclusions made in the paper.

4. How about the polymer molecular weight difference by increasing the glycol sidechain content? It is also important for photophysical properties in solutions and solid films and should be discussed.

Our response: We thank the reviewer for their comment and for highlighting the importance of the molecular weight of the conjugated polymers. As we responded to reviewer 1's comment 2, the initial GPC measurements were conducted at room temperature in chloroform solvent and yielded large PDIs for many of the polymers. This likely occurred due to aggregation effects because of the low temperature that the measurements were conducted in. To ensure an accurate measure of the molecular weight given its importance to our conclusions, we have enrolled the help of specialists in the GPC of polymers to remeasure the molecular weights. GPC measurements were done at 40°C in chloroform, the elevated temperature improves reliability of the measurement by reducing aggregation effects. The results are summarised in the table below and show that the g0%, g25%, g50% and g100% polymers have similar molecular weights. We can therefore be confident that the results of the photophysical properties in solution and thin film presented, and conclusions drawn throughout the manuscript do not arise from molecular weight differences, but from the effect of the glycol sidechain content.

Table R1: Summary of the Mn and Mw values for the CPDT polymer series, measured with GPC at 40°C in chloroform.

Polymer	Mn (kDa)	Mw (kDa)	PDI
g0%	22	49	2.2
g25%	22	50	2.3
g50%	23	55	2.4
g75%	13	24	1.8
g100%	20	66	3.4

We have updated the polymer molecular weights in the supplementary information, section 1 (Experimental Synthesis), which are highlighted in the text (Lines 148-179), and added a summary table of the measurements to section 1 (Line 251):

Table S1: Summary of the Mn and Mw values for the CPDT polymer series, measured with GPC at 40°C in chloroform.

Polymer	Mn (kDa)	Mw (kDa)	PDI
g0%	22	49	2.2
g25%	22	50	2.3
g50%	23	55	2.4
g100%	20	66	3.4

We have added Adam Marsh as a co-author on the paper as he carried out the GPC measurements. Line 4 and 17:

Adam V. Marsh,⁵

⁵ Physical Science and Engineering Division, KAUST Solar Center (KSC), King Abdullah University of Science and Technology (KAUST), Thuwal 23955-6900, Saudi Arabia.

5. In the part of steady-state Raman spectroscopy, the author only discussed the difference in peak intensity for polymers. How about the effect of the variation of peak shape?

Our response: Figure 3 shows the normalised ground state Raman spectra for the CPDT polymers. Relative peak intensities provide information on π -electron density to a given bond in the backbone. Peak shape is influenced by packing structure and order/disorder in the polymer films, with a broader peak being indicative of a more disordered packing structure due to a wider range of effective conjugation lengths being present in the film.⁴ For the data shown in Figure 3(b), there is little to no variation in peak shapes and any apparent differences in peak shape is due to changes in relative intensity of the peaks. To clarify this, the normalised Raman mode has been labelled in Figure 3 (Line 278) and peaks 1-3 relabelled as peaks 1-4 in the figure and throughout the text (Line 251):

Peak 1 at 1370cm^{-1} , the shoulder of the main Raman peak at around 1380cm^{-1} , is assigned to the C-C intra-ring mode on the central cyclopentane ring of the CPDT unit. Peak 3 at around 1420 cm^{-1} is assigned to the C-C intra-ring mode on the thiophene rings. Peak 4 at around 1520cm^{-1} is assigned to the outer C=C mode on the thiophene rings and the neighbouring C-C inter-ring mode between the CPDT units.

Figure 3: (a) Raman peak assignments of the two vibrational modes of interest. Peak 1, at around 1370 cm⁻¹, corresponds to the C-C intra mode on the cyclopentane ring. Peak 4, at around 1520 cm⁻¹, corresponds to the C=C intra thiophene mode, and the neighbouring C-C inter mode between the CPDT units. (b) Raman spectra for the CPDT polymers, normalised to the 1380 cm⁻¹ mode peak 2*, thin films on prefabricated Fraunhofer OFET substrates, using 785 nm excitation wavelength. (c) Relative peak intensities of peaks 1 and 4 for the CPDT polymers.

For increasing glycol content, there is an increase in relative intensity of peak 1 and a decrease in peaks 3 and 4. There is also a slight downshift in peak 4 above g25%, which is indicative of an increase in bond length, which arises due to a loss of π -electron density from the thiophene C=C intra and C-C inter bonds as it becomes more localised to the centre of the CPDT unit. This is accompanied by a slight broadening of the peak, which is indicative of a more disordered packing structure in the higher glycol content films. These are the main changes in the Raman spectra seen with increasing glycol content and the text in the manuscript has been revised to emphasise this and highlight changes in peak shape (Line 262):

In addition to the relative peak intensity changes seen in the spectra, peak 4 shows a slight downshift for glycol contents above g25%. This is indicative of an increase in bond length, which arises due to a loss of π -electron density from the thiophene C=C intra and C-C inter bonds as it becomes more localised to the centre of the CPDT unit. A slight broadening of peak 4 can also be seen, which is reflective of the more disordered packing structure in the higher glycol content films.

6. It would be more convincing to elaborate on the localization of π -electron density in the backbone by cooperating with theory calculation or other characterization.

Our response: We thank the reviewer for this interesting suggestion. To support the conclusions made experimentally regarding a more localised π -electron density in the ground state g100% polymer backbone, we carried out further DFT calculations to simulate the electrostatic potential distribution (ESP) along the backbone of the ground state g0% and g100% trimers. This provides

further insight into the impact of the glycol sidechains on the intrinsic, intramolecular π -electron distribution along the conjugated backbone. The ESPs of the g0% and g100% trimers were simulated according to the Merz-Kollman (MK) scheme and ESP was then calculated for each bond along the backbone and averaged according to bond type across the trimers. SI Figure S34 has been added, which shows the average ESP for each bond type along the backbone for the g0% and g100% trimers in the ground state and the difference in the average ESP for each bond upon glycolation of the trimer. The results show that compared to the alkylated trimer, the glycolated trimer has less π -electron density localised to the external thiophene rings and more π -electron density localised to the internal cyclopentane ring. This is consistent with the experimental results shown in Figure 3 and supports the conclusion of π -electron density being more localised to the central cyclopentane ring in the g100% polymer. The main manuscript text has been revised to reference these DFT simulations (Line 269):

To further support this, DFT was used to simulate the electrostatic potential distribution (ESP) according to the Merz-Kollman (MK) scheme⁴⁵ along the backbone of ground state g0% and g100% trimers (Figure S34). The results show that compared to the g0% trimer, the g100% trimer has less π -electron density localised to the outer thiophene rings, and more π -electron density localised to the central cyclopentane ring. This is consistent with the experimental results shown in Figure 3 and supports the conclusion of the π -electron density being more localised to the centre of the CPDT unit in the g100% polymer.

The supplementary information has been revised to include the ESP figure and explanatory text (Line 390):

Figure S34: (a) Bond labels for the g0% and g100% oligomers. (b) Average ESP for g0% and g100% trimers for each bond along the backbone and C-C sidechain, being the first carbon-carbon bond in the sidechain. (c) Difference in average ESP between the g0% and g100% trimers, showing the impact of full glycolation on the ESP along the backbone.

To support the conclusions made experimentally regarding π -electron density being more localised to the cyclopentane ring in the ground state g100% polymer backbone, we carried out further DFT calculations to simulate the electrostatic potential distribution (ESP) along the backbone of the ground state g0% and g100% trimers. This provides further insight into the impact of the glycol sidechains on the intrinsic, intramolecular π -electron distribution along the conjugated backbone. The ESPs of the g0% and g100% trimers were simulated according to the Merz-Kollman (MK) scheme and ESP was then calculated for each bond along the backbone and averaged according to bond type across the trimers. Figure S34 shows the average ESP for each bond along the backbone for the g0% and g100% trimers in the ground state and the change in the average ESP for each bond upon glycolation of the trimer. The results show that compared to the alkylated trimer, the glycolated trimer has less π -electron density localised to the outer thiophene ring C=C_{inner} and C-C_{intra1} bonds and more π -electron density localised to the central cyclopentane ring C-C_{intra2} and C-C sidechain bonds. So, the g100% trimer has greater π -electron density concentration in the internal ring of the CPDT unit. This is consistent with the experimental results shown in Figure 3 and supports the conclusion of a more localised π -electron density in the g100% polymer.

7. From kinetic analysis, how could we distinguish the effect between disorder and polaron formation on faster singlet decay and photoluminescence quenching in g100% polymer thin film?

Our response: We thank the reviewer for raising an interesting point. It is difficult to fully distinguish between the impact of disorder and enhanced polaron pair formation on the faster singlet signal decay and PL quenching seen for the g100% polymer thin film. The best way to gain insight into the processes' relative contributions would be to compare the time constants of the TAS singlet signal decay and polaron pair signal rise in the kinetic analysis. If enhanced polaron pair formation were the only factor causing faster singlet decay and PL quenching, the time constants for singlet signal decay and polaron pair signal rise would match well.

This comparison of time constants, however, does not account for the fact that disorder will also contribute to the enhanced polaron pair formation seen. In the manuscript we conclude that there is enhanced polaron pair formation in the g100% polymer due to its higher reorganisation energy, which allows longer time for structural relaxation and hence enhances vibronic coupling to the polaron pair state. The disorder induced by the glycol sidechains will facilitate recombination of singlet excitons and hence quench the singlet TAS signal and PL intensity, as is stated on line 405 of the manuscript: 'The faster geminate recombination may be due to the more disordered morphology of the g100% polymer, as outlined in section 2.1. This will likely inhibit delocalisation of the PPs and hence facilitate faster recombination.^{60,61}'. It will also, however, contribute to the higher reorganisation energy seen for the g100% polymer via the intermolecular component, which in turn facilitates polaron pair formation.

So, as the two effects are interlinked it is difficult to make a comment on or quantify their contributions to singlet decay and PL quenching. The main point of interest in the TAS data is that there is enhanced polaron pair formation in the glycolated polymer, induced by a higher reorganisation energy. So, whereas it may be interesting to comment on the relative contributions of disorder and enhanced polaron pair formation to singlet signal decay and PL quenching, we believe it to not be critical for supporting the conclusions we are making here.

8. More description and explanations should be added in the supporting information for the second part materials properties (Figure S25-S27, and Table S1).

Our response: We thank the reviewer for pointing this out and have revised the SI accordingly.

More text has been added referencing Figures S25-S27 (Line 259):

Solution and film UV-Vis-NIR absorption spectra were recorded by using a Cary 5000 UV-Vis-near IR double-beam spectrophotometer and are shown in Figure S25. Cyclic voltammetry (CV) was performed at a room temperature in a 0.1 M solution of tetrabutylammonium perchlorate in acetonitrile under nitrogen gas at a scan rate of 50 mV/s. A glassy carbon electrode as the working electrode, a Pt wire was used as the counter electrode and an Ag/AgCl electrode as the reference electrode. The CV scans of the CPDT polymers are shown in Figure S26. Figure S27 shows the thermogravimetry analysis (TGA) and differential scanning calorimetry (DSC) curves of the CPDT polymers. As of glycol sidechain content increases, T_d and T_m gradually decrease while crystallinity distinctly appears with T_c and T_m .

The figure captions of Figures S25-S27 have also been revised (Line 281, 296 and 330):

Figure S25: Solution and thin-film UV-vis absorption spectra of the CPDT polymers. For both solution and thin film measurements chloroform was used as the solvent.

Figure S26: Cyclic voltammogram (CV) curves of the CPDT polymers.

Figure S27: Thermogravimetry Analysis (TGA) and differential scanning calorimetry (DSC) curves of the CPDT polymers.

Information has been added to Table S1 (Line 338 and 339):

d) HOMO level was calculated from oxidation onset.

e) LUMO level was obtained from HOMO and E_g .

9. Format and details of supporting information should be revised.

Our response: Formatting and details of the supporting information have been revised as detailed in the response above.

Reviewer #3 (Remarks to the Author):

Reviewer #4 (Remarks to the Author):

The paper by Pagano and coauthors describes the effect of polymer glycolation (when alkyl sidechains are replaced by glycol sidechains) on the optoelectronic properties of a series of cyclopentadithiophene (CPDT) conjugated polymers. They observe an enhanced polaron formation in polymers with larger glycol sidechain content. While such an effect is commonly attributed to an enhanced dielectric constant, the authors here attribute it to higher reorganization energy and enhanced vibrational coupling between singlet excitons and polaron pair states. This is the key result of the paper. If correct, then this is an important conclusion.

The observation of an enhanced polaron formation upon glycolation is fully supported by their experiments. The authors performed an impressive set of measurements in a detailed and careful manner, including ultrafast transient absorption spectroscopy and femtosecond stimulated Raman spectroscopy, to monitor the structural relaxation dynamics of their polymers upon photoexcitation. Their arguments to correlate the enhanced polaron formation with enhanced vibrational coupling are plausible. I am, however, not sure whether I am at present fully convinced by the ir assignment to reorganization energy, though this is a point that may be addressed in a revised version. My general opinion is that the paper describes a very interesting study which is of considerable interest to the field, hence I recommend publication once the following points have been addressed.

Our response: We thank the reviewer for their time taken to review our manuscript and their recognition of the important experimental results of our work, which highlights the impact of reorganisation energy on polaron pair formation.

1. To support their correlation between increased polaron pair formation and reorganization energy, the authors quote (in Fig. 2c) the Stokes' shift and Huang-Rhys parameter, as well as calculations for oligomers up to length of 5 units. The calculations indeed show larger reorganization energies for the glycolated compounds, yet straight line fits through both sets of data point indicate that the energies converge for oligomers longer than about 7 repeat unit, so this only holds if the conjugation length is sufficiently short. The Stokes' shift is read out from Fig. S30. This is a central argument and therefore this figure should be placed in the main manuscript. It would also help if it was plotted on an energy scale and with a range that actually allows reading off values. At present – on a wavelength scale from 500 to 800 nm – it is not possible to judge the validity of the data presented in Figure 2c (the distances between the 0-0 peaks differ by less than 1 mm in a A4 printout). Similarly, I am doubtful about the Huang-Rhys parameter data. In different solvents, the spectra have different linewidth, and the 0-0 and 0-1 peak overlap. So just taking the ratio between them may give erroneous values. It may be that their data supports the conclusions, yet from the current presentation of the data it is not possible to assess this. It would help if the authors could decompose the spectra at least into a sum of Gaussian peaks with equal width to clearly identify the 0-0 positions of absorption and emission, and the correct height of the 0-0 and 0-1 peak. When doing this, it is important to plot the reduced absorption and emission spectra, to correctly identify the geometric reorganization energy (as half of the distance between the two 0-0 peaks). A full Franck-Condon Analysis would, of course, be even better, in particular since the Raman data for the materials are available.

Our response: We thank the reviewer for their points and for their suggestions on how to improve the presentation of the data.

Regarding the reorganisation energies converging at large oligomer lengths, while the data show that the differences in reorganisation energy are largest at shorter oligomer lengths, these short effective conjugation lengths are still relevant to the solid-state thin film systems that we have studied. The absorption and GIWAXS data show that packing in both the g0% and g100% films is not highly ordered, which means that the average conjugation length is likely to be low. This will be exacerbated in the g100% film, which has a more disordered packing structure relative to the g0% film. As the computational simulations are intended as guidelines, we believe that they are sufficient to support the experimental data that shows clear evidence of reorganisation energy differences in the Stoke's shift and Huang-Rhys factor values (see below).

We have increased the detail of the labelling on Figure S30 and changed the x-axis to energy. Line 354 in the SI:

Figure S30: Normalised absorbance and PL spectra of the g0% (top), g50% (middle) and g100% (bottom) polymers in solution. Spectra were taken in toluene, chlorobenzene (CB), chloroform (CF) and dichloromethane (DCM), and polymers were at a concentration of 0.01mg/ml. A shoulder of emission can be seen in the g0% DCM absorption spectrum, which likely arises due to the poor solubility of the alkylated polymers in the polar solvent.

The extracted Stoke's shift values are plotted as a function of glycol percentage in Figure 2(c) so that they can be compared with the H-R factors and calculated reorganisation energies without dominating Figure 2. We think this option is best to concisely convey the important information, while including the full data sets in the SI is sufficient to ensure the detail is there and to support our extracted data.

As per the reviewer's suggestion, we have decomposed the solution absorption and emission spectra into a sum of Gaussian peaks with equal width and spacing to identify the heights of the 0-0 and 0-1 peaks in the absorption spectra to extract HR factors and identify the 0-0 positions to extract Stokes shift values. Up to three Gaussian peaks of equal FWHM and spacing provided sufficient fitting to the data, see an example for the g0% polymer in DCM Figure R4 and Figure R5 below.

Figure R4: (a) An example of the Gaussian fitting done to the solution absorption data, with g0% in DCM. The experimental spectrum was decomposed into three Gaussian peaks with equal width and spacing and gives a sufficient fit, allowing us to extract the 0-0 and 0-1 peak positions and intensities. Details of the peak fittings are shown in (b).

Figure R5: (a) An example of the Gaussian fitting done to the solution PL data, with g0% in DCM. The experimental spectrum was decomposed into three Gaussian peaks with equal width and spacing and gives a sufficient fit, allowing us to extract the 0-0 peak position. Details of the peak fittings are shown in (b).

These examples have been added to the SI (Line 365):

An example for the g0% polymer in DCM is shown in Figure S31 Figure S32 below.

(b)

Peak Index	Peak Position (eV)	FWHM	Max Height
1	1.96	0.18	0.85
2	2.12	0.18	0.72
3	2.27	0.18	0.50

Figure S31: (a) An example of the Gaussian fitting done to the solution absorption data, with g0% in DCM. The experimental spectrum was decomposed into three Gaussian peaks with equal width and spacing and gives a sufficient fit, allowing us to extract the 0-0 and 0-1 peak positions and intensities. Details of the peak fittings are shown in (b).

(b)

Peak Index	Peak Position (eV)	FWHM	Max Height
1	1.86	0.10	0.97
2	1.74	0.10	0.39
3	1.65	0.10	0.28

Figure S32: (a) An example of the Gaussian fitting done to the solution PL data, with g0% in DCM. The experimental spectrum was decomposed into three Gaussian peaks with equal width and spacing and gives a sufficient fit, allowing us to extract the 0-0 peak position. Details of the peak fittings are shown in (b).

New HR-factors and Stoke's shift values were extracted from the fittings (see below) and are similar to our previous results. The values obtained show the same trend with increasing glycol content and thus does not differ largely from our previous data analysis or the conclusions drawn.

The main manuscript has been revised to describe the fittings done to the data when HR factor and Stoke's shift calculations are introduced (Line 227):

To accurately extract the positions and intensities of the 0-0 and 0-1 peaks from the solution absorption and PL spectra, Gaussian fittings were applied to the data (see SI for details).

The details of the fittings are described in the SI (Line 358):

For HR-factor and Stoke's shift calculations, the solution absorption and PL spectra shown in Figure S30 were deconvoluted into a sum of Gaussian peaks of equal width and spacing in order to accurately identify and extract the positions and intensities of the 0-0 and 0-1 vibronic peaks. Up to three Gaussian peaks of equal FWHM and spacing provided sufficient fitting to the data, with the peak spacing roughly equal to the energy of the highest intensity ground state Raman mode.

Figure 2 in the main manuscript has been revised to include the newly calculated HR-factors and Stokes shift values (Line 187):

Figure 2: (a) Schematic of the PESs for the ground (S_0) and excited (S_1) states showing the vertical transitions that occur during the electronic transitions, the PES minima displacement (ΔQ), and the reorganisation energies associated with vibrational relaxation to the PES minimum after an electronic transition occurs. $E_{g_0}^0$ is the energy of the ground state molecule at the optimised geometry of the ground state molecule, $E_{g_1}^0$ is the energy of the ES molecule at the optimised geometry of the ES molecule and $E_{g_0}^1$ is the energy of the ground state molecule at the optimised geometry of the ES molecule. λ_1 and λ_2 are the reorganisation energies associated with IVR on the S_1 and S_0 PESs, respectively. (b) DFT-calculated singlet λ values for a monomer, trimer and pentamer of the $g0\%$ and $g100\%$ polymers. (c) Stokes shift values (top) taken as the energy difference between the positions of the 0-0 peak of the solution absorption and PL spectra. Huang-Rhys Factors (bottom) calculated as the ratio of the 1-0/0-0 solution absorption peak intensities.

We take the excited-state geometric reorganisation energy (RE) to be half of the Stoke's shift (SS) and compare them to the DFT-calculated excited-state reorganisation energy values, which are defined in the Experimental Methods section as $\lambda_1 = E^1g_0 - E^1g_1$, in the table below. The simulated values are larger than the halved experimental Stoke's shift values, which is due to limitations to shorter oligomer lengths in the simulations. Despite this discrepancy between the values, the trend with increasing glycol percentage is consistent.

Table R2: DFT-calculated λ_1 values and halved Stoke's shift values for each solvent for the g0% and g100% polymers.

Polymer	DFT-calculated excited-state RE, λ_1 , (eV)	$\frac{SS}{2}$ (eV)			
		Toluene	Chlorobenzene	Chloroform	DCM
g0%	0.18	0.053	0.057	0.060	0.049
g100%	0.24	0.062	0.064	0.065	0.069

2. The assignment to vibrational coupling is based on the time resolved TAS and Raman spectra, and this seems convincing. My only point of critique relates to the relative position of S1 and the PP state in Figure 6. The authors present data for a single compound in solution. For a single compound, the charge-transfer state that forms upon transferring an electron to an adjacent chromophore is usually at higher energy than the singlet state where it originates from, due to the coulomb attraction that needs to be overcome when separating out electron and hole (and that is much larger than any reduction due to additional reorganization energy). This is currently not reflected in Figure 6, where the PP minimum is shown below the S1 state. Perhaps this could be corrected.

Our response: While the reviewer raises a valid point regarding the relative positions of the S₁ and PP state minima for a single compound in solution, the data we present is not for solutions but for thin films of the g0% and g100% polymers. This is highlighted in the caption of Figure 4 (line 328, 'Raw TA spectra for thin film samples of ...') and on line 386 ('Enhanced PP formation in the higher glycol content films...').

To clarify this, we have added text to line 315:

TAS measurements of **thin films of** the g0% and g100% polymers were carried out...

Line 410:

FSRS spectra of the g0% and g100% polymers **in thin films** were collected and are shown in Figure 5(a).

And have updated the figure captions of Figure 5 (Line 417) and Figure 6 (Line 522):

Figure 5: (a) FSRS Raman gain spectra of the g0% (top) and g100% (bottom) polymers **in thin films**. Negative peaks at $\sim 1390\text{ cm}^{-1}$ and $\sim 1550\text{ cm}^{-1}$ are indicative of ground state bleaching of the polymer upon photoexcitation. The positive peak at $\sim 1350\text{ cm}^{-1}$ is indicative of ES formation. The two main modes of interest, 1390 cm^{-1} and 1350 cm^{-1} are labelled GSB (ground state bleach) and ES (excited state), respectively. (b) Time-resolved normalised Raman gain intensity of the 1390 cm^{-1} GSB (top), 1350 cm^{-1} ES (middle) modes and time-resolved peak shifts of the 1350 cm^{-1} ES mode (bottom). Vertical dashed lines indicate when the maximum relative Raman gain intensities are reached and when ES peak shifting stops for each polymer.

Figure 6: PES schematic showing the processes of vibrational cooling (thermalisation) and polaron pair state formation in thin film of (left) the g0% polymer, which thermalises in 0.6 ps to the minimum of the S_1 state and subsequently forms polaron pairs (labelled PP), thermalising to the minimum of the PP state by 1.25 ps and (right) the g100% polymer, where vibrational coupling facilitates PP formation and the polymer thermalises in 5 ps to the minimum of the PP state. Green arrows indicate measured vibrational cooling times. ΔQ is the offset of the S_0 and S_1 PESs and is larger for the g100% polymer, as shown with the solvatochromic measurements. $k_{r,0}$ and $k_{r,100}$ are the radiative recombination rates for the g0% and g100% polymers, respectively. The PESs of the g100% polymer are shown to be broader than those of the g0% polymer, which is a result of the higher structural disorder in the polymer.^{40,63}

In Figure 6, we are describing polaron pair state formation in a solid-state system, not in single compounds in solution. On line 349, we define polaron pair states as ‘excitons with increased charge transfer character, with the electron-hole pair being spatially separated and bound with a weaker Coulomb attraction.’ Importantly, the polaron pairs are not charge separated states, the electron and hole are still Coulombically bound, and their formation is induced via vibrational coupling from the S_1 state, which acts to stabilise the state. So, there is no hopping process of the electron between adjacent chromophores. As this is a thin film homopolymer system, the polaron pair state is predominantly intermolecular in nature, being delocalised between packed polymer chains.⁵ The spatially delocalised, interchain nature of the state means that the minimum of the polaron pair state is below the minimum of the S_1 state in Figure 6. This is similar to how a CT-state in a donor-acceptor blend system is stabilised relative to the S_1 state.⁶

A sentence has been added to the main manuscript to make this clearer when polaron pairs are first introduced (Line 351):

As this is a thin film homopolymer system, the PP state is likely predominantly intermolecular in nature, being delocalised between packed polymer chains.⁵⁴ This stabilises the state relative to the singlet exciton, similar to how a CT-state in a donor-acceptor blend system is stabilised relative to the singlet state.⁴

Minor issues:

3. Line 157: The authors claim that “ λ is the energy required for the molecular lattice to distort upon ES formation”, however, in fact only λ_1 is required for that. The same comment is applicable to Eq.(1) – only λ_1 energy is needed to form ES. The authors may wish to correct this.

Our response: We thank the reviewer for bringing this to our attention and have rectified the text to specify that λ is the energy required for both excited state formation and decay (Line 172):

As outlined in the introduction, λ is the energy required for the molecular lattice to distort upon ES formation and decay.

In equation 1, we want to describe relaxation to the PES minimum of both the ground and excited states to describe the total inner reorganisation energy. We have made this clearer in the text (Line 197 and Line 201):

The energy change associated with nuclear relaxations to the minima of the ground and excited-state PESs is the intramolecular (inner) λ , λ_i :

Where λ_1 and λ_2 are the excited-state and ground-state reorganisation energies, respectively.

4. Lines 490-491: A reference to Fig.5(b), by which the values of 0.6nm and 1.25ns were obtained, would be helpful.

Our response: A reference to Figure 5(b) has been added on line 514:

For the g0% polymer, the structural relaxation to the minimum of the S_1 state occurs in 0.6 ps. This is followed by PP formation, after which there is vibrational cooling until 1.25 ps (blue and green vertical dashed lines in Figure 5(b)).

5. In Fig. 4 and 5, longer ticks would be very helpful to read off values (in particular for the log scale in Figure 4b).

Our response: The figures have been revised with longer ticks, see below (Lines 328 and 417).

Figure 4: Raw TA spectra for thin film samples of the (a) g0% and (c) g100% polymers at different pump-probe delay times, probed from 630 nm-1060 nm after photoexcitation at 600 nm, $17\mu\text{J}/\text{cm}^2$. Spectra are averaged over the delay times indicated. The negative band corresponds to ground-state bleach (640 nm) and stimulated emission from the excited-state (720 nm). The broad, positive band in the NIR region is attributed to the photoinduced absorption of the photoexcited state. The shorter-lived signal centred at ~ 1050 nm is attributed to singlet excitons (labelled 'S') and the longer-lived signal centred at ~ 980 nm to polaron pairs (labelled 'PP'). (b) Normalised kinetics of the deconvoluted singlet TA spectrum, fitted with a biexponential decay (dashed lines). (d) Normalised kinetics of the deconvoluted PP TA spectrum, fitted with a multi-exponential (dashed lines).

Figure 5: (a) FSRS Raman gain spectra of the g0% (top) and g100% (bottom) polymers **in thin films**. Negative peaks at ~ 1390 cm^{-1} and ~ 1550 cm^{-1} are indicative of ground state bleaching of the polymer upon photoexcitation. The positive peak at ~ 1350 cm^{-1} is indicative of ES formation. The two main modes of interest, 1390 cm^{-1} and 1350 cm^{-1} are labelled GSB (ground state bleach) and ES (excited state), respectively. (b) Time-resolved normalised Raman gain intensity of the 1390 cm^{-1} GSB (top), 1350 cm^{-1} ES (middle) modes and time-resolved peak shifts of the 1350 cm^{-1} ES mode (bottom). Vertical dashed lines indicate when the maximum relative Raman gain intensities are reached and when ES peak shifting stops for each polymer.

6. Line 340: What does GA stand for?

Our response: We thank the reviewer for bringing this to our attention and have defined the term on line 361:

To further understand the ES dynamics, we apply global analysis (GA) to deconvolute the kinetics of each ES species in the TA spectra.

Reviewer #5 (Remarks to the Author):

References

1. Stewart, K. et al. Understanding Effects of Alkyl Side-Chain Density on Polaron Formation Via Electrochemical Doping in Thiophene Polymers. *Adv. Mater.* 2211184 (2023).
2. Mohapatra, A. A. et al. Rational Design of Donor-Acceptor Based Semiconducting Copolymers with High Dielectric Constants. *Journal of Physical Chemistry C* **125**, 6886–6896 (2021).
3. Torabi, S. et al. Strategy for enhancing the dielectric constant of organic semiconductors without sacrificing charge carrier mobility and solubility. *Advanced Functional Materials* **25**, 150–157 (2015).
4. Tsoi, W. C. et al. The nature of in-plane skeleton Raman modes of P3HT and their correlation to the degree of molecular order in P3HT:PCBM blend thin films. *J. Am. Chem. Soc.* **133**, 9834–9843 (2011).
5. Reid, O. G., Pensack, R. D., Song, Y., Scholes, G. D. & Rumbles, G. Charge photogeneration in neat conjugated polymers. *Chem. Mater.* **26**, 561–575 (2014).
6. Clarke, T. M. & Durrant, J. R. Charge photogeneration in organic solar cells. *Chem. Rev.* **110**, 6736–6767 (2010).

REVIEWER COMMENTS

Reviewer #1 (Remarks to the Author):

The authors have addressed all my comments properly. Thus, I recommend the paper to be published as is.

Reviewer #2 (Remarks to the Author):

Manipulating the excitonic nature of organic semiconductors is still a major challenge. On the one hand, stable tunable exciton energies are desired to optimize OLEDs. On the other hand, controlling the lifetime of the excited state is understood as one powerful strategy to decide on the fate of the excited state as required for the optimization of solar cells. The paper by Pagano and coauthors describes the effect of polymer glycolation on a series of cyclopentadithiophene (CPDT) conjugated polymers. One key observation is the identification of a slightly favoured polaron formation for polymers with a higher content of glycol side-chains. The current literature typically discusses enhanced polaron formation in conjugated materials with polar side chains in terms of epsilon and increased polarization. Here, the authors present experimental and theoretical evidences that the higher reorganization energy of glycol substituted CPDT is the root cause for an enhanced polar formation. This is assisted by enhanced vibrational coupling between singlet excitons and the polarons. Although the variations in formation energy and reaction kinetics reported are rather small, this finding is of utmost importance, as it opens up an excitingly new opportunity to manipulate the excited state in conjugated semiconductors. Being able to manipulate the reorganization energy independently from polaron formation has the potential to balance transport and excited state dynamics. Here, the authors provide convincing evidence that this can be achieved almost "seamless" by the variation of the percentages of glycol sidechains. Glycol substituted conjugated donor and acceptor materials have not become a mainstream research direction yet, though they were frequently discussed as a strategy to reduce the exciton binding energy of fullerenes, NFAs or donor-type conjugated polymer. That was to far impeded by the risk that a larger dielectric constant may enhance charge generation in parallel to reducing charge carrier transport. In summary, Pagano et al present strong evidence that polar sidechains not only change the effective dielectric constant but as well impact the reorganization energy for photoinduced excited state formation. The revised version is sensitively discussing their model as a function of glycol sidechain percentage and properly address the influence of molecular weight variations. The vibronic coupling between the excitonic and the polaronic state is sound and should be followed for other materials with more complex donor-acceptor character.

Overall, the paper fulfills the requirements for publication in Nat. Comm..

Reviewer #3 (Remarks to the Author):

Reviewer #4 (Remarks to the Author):

I have considered the authors' rebuttal letter. I am afraid, I feel that our points 1 and 2 have not entirely been addressed adequately (the technical issues 3-6 have been resolved satisfactorily). I like

the data and still feel it merits publication, but I am not convinced by the already mentioned shortcomings in their analysis and interpretation, and in this respect the manuscript currently falls short of the standard expected by a Nature Communications paper. Let me illustrate this.

Regarding point 1 concerning the analysis of the reorganization energy. A central claim of the paper is that a larger reorganization energy in glycolated polymers would assist polaron formation. I still have serious reservations on this. To make the argument more convincing, we suggested that the authors may undertake a careful analysis of the reorganization energy through a Franck-Condon analysis. The attempt presented in the rebuttal letter is not convincing. (i) the authors did not use the reduced absorption and absorption spectra (number of photons per energy interval), which is important for correct determination of vibrational peak heights, (ii) the spacing used for the Gaussian peaks is not equal, and does not correspond to the energies of the dominant Raman modes, (iii) the authors present their analysis only for the non-glycolated polymer, yet not for the glycolated one (and only for a small range of the spectra). This prevents a reader from forming his own opinion on the validity of their approach.

Regarding point 2 concerning the presentation of figure 6. Here, the authors further comments helped to clarify my questions. If I understand correctly, the authors place the PP state at lower energy than the S1 state as they consider the delocalisation in an excimer (and the associated stabilization) overcompensates the destabilization resulting from a larger spacing. This is possible, yet it also raises the question whether the difference in PP formation between the glycolated and non-glycolated polymer may rather be a result from the different morphologies and associated excimer content than a result from the differences in reorganization energy (the experimentally derived reorganization energies differ by only 10 meV, i.e. less than kT , which makes it hard to believe that this should make much difference). A critical evaluation of this aspect, e.g. in the discussion, might be worthwhile. I do not wish to render publication of this manuscript difficult. I have detailed here my concerns so that the authors have a chance to consider them, assuming they also wish for their work to be presented in a way accessible and convincing to many readers.

Reviewer #5 (Remarks to the Author):

I co-reviewed this manuscript with one of the reviewers who provided the listed reports as part of the Nature Communications initiative to facilitate training in peer review and appropriate recognition for co-reviewers

Slow Vibrational Relaxation Drives Ultrafast Formation of Photoexcited Polaron Pair States in Glycolated Conjugated Polymers

Manuscript: NCOMMS-23-34380A

Responses to the reviewers' comments

We thank all reviewers for their time to read the manuscript and for their encouraging and constructive comments. We believe that we have addressed all of the reviewers' comments. Below are our detailed replies to the reviewers' comments and questions. Our detailed point by point responses are in blue and the changes in the manuscript are highlighted in cyan. We hope that our revised manuscript is now acceptable for publication in *Nature Communications*.

Reviewer #1 (Remarks to the Author):

The authors have addressed all my comments properly. Thus, I recommend the paper to be published as is.

Our response: We thank the reviewer for their comments and contribution.

Reviewer #2 (Remarks to the Author):

Manipulating the excitonic nature of organic semiconductors is still a major challenge. On the one hand, stable tunable exciton energies are desired to optimize OLEDs. On the other hand, controlling the lifetime of the excited state is understood as one powerful strategy to decide on the fate of the excited state as required for the optimization of solar cells. The paper by Pagano and coauthors describes the effect of polymer glycolation on a series of cyclopentadithiophene (CPDT) conjugated polymers. One key observation is the identification of a slightly favoured polaron formation for polymers with a higher content of glycol side-chains. The current literature typically discusses enhanced polaron formation in conjugated materials with polar side chains in terms of epsilon and increased polarization. Here, the authors present experimental and theoretical evidences that the higher reorganization energy of glycol substituted CPDT is the root cause for an enhanced polar formation. This is assisted by enhanced vibrational coupling between singlet excitons and the polarons. Although the variations in formation energy and reaction kinetics reported are rather small, this finding is of utmost importance, as it opens up an excitingly new opportunity to manipulate the excited state in conjugated semiconductors. Being able to manipulate the reorganization energy independently from polaron formation has the potential to balance transport and excited state dynamics. Here, the authors provide convincing evidence that this can be achieved almost "seamless" by the variation of the percentages of glycol sidechains. Glycol substituted conjugated donor and acceptor materials have not become a mainstream research direction yet, though they were frequently discussed as a strategy to reduce the exciton binding energy of fullerenes, NFAs or donor-type conjugated polymer. That was to far impeded by the risk that a larger dielectric constant may enhance charge generation in parallel to reducing charge carrier transport. In summary, Pagano et al present strong evidence that polar sidechains not only change the effective dielectric constant but as well impact the reorganization energy for photoinduced excited state formation. The revised version is sensitively discussing their model as a function of glycol sidechain percentage and properly address the influence of molecular weight variations. The

vibronic coupling between the excitonic and the polaronic state is sound and should be followed for other materials with more complex donor-acceptor character. Overall, the paper fulfils the requirements for publication in Nat. Comm.

Our response: We thank the reviewer for their very positive and encouraging comments.

Reviewer #3 (Remarks to the Author):

Reviewer #4 (Remarks to the Author):

I have considered the authors' rebuttal letter. I am afraid, I feel that our points 1 and 2 have not entirely been addressed adequately (the technical issues 3-6 have been resolved satisfactorily). I like the data and still feel it merits publication, but I am not convinced by the already mentioned shortcomings in their analysis and interpretation, and in this respect the manuscript currently falls short of the standard expected by a Nature Communications paper. Let me illustrate this. Regarding point 1 concerning the analysis of the reorganization energy. A central claim of the paper is that a larger reorganization energy in glycolated polymers would assist polaron formation. I still have serious reservations on this. To make the argument more convincing, we suggested that the authors may undertake a careful analysis of the reorganization energy through a Franck-Condon analysis. The attempt presented in the rebuttal letter is not convincing. (i) the authors did not use the reduced absorption and absorption spectra (number of photons per energy interval), which is important for correct determination of vibrational peak heights, (ii) the spacing used for the Gaussian peaks is not equal, and does not correspond to the energies of the dominant Raman modes, (iii) the authors present their analysis only for the non-glycolated polymer, yet not for the glycolated one (and only for a small range of the spectra). This prevents a reader from forming his own opinion on the validity of their approach.

Our response: We thank the reviewer for their comments and guidance on our fitting methods. Extraction of exact Huang-Rhys factors is not the main focus of this work and is instead used to provide additional confirmation of an increasing reorganisation energy with increasing glycol content, which was shown in our DFT simulations. Although detailed Franck-Condon analysis is beyond the scope of this work, we have considered the fitting again and now account for there being specific vibrational modes coupling to the optical transition. These are the 1380 cm^{-1} and 1520 cm^{-1} modes, corresponding to peak 1 and peak 4 in the steady-state Raman spectra, respectively. Peaks 1 and 4 show the largest changes upon increasing glycol content and also show large ground-state bleaching in the FSRS spectra, therefore contributing most to the structural change upon photoexcitation as expected. We therefore fit the solution absorption spectra with a sum of two sets of gaussian peaks, with the peak spacing in one set corresponding to the energy of the 1380 cm^{-1} mode (0.17 eV) and the peak spacing in the other set corresponding to the energy of the 1520 cm^{-1} mode (0.19 eV). Examples for the g0% and g100% absorption spectra in chlorobenzene are shown in Figures R1-R4 below:

Figure R1: Gaussian fitting of the g0% absorption spectrum in chlorobenzene. The experimental data (black crosses) were fit with the sum of two sets of gaussian peaks, each with a unique, fixed spacing corresponding to a mode in the steady-state Raman spectra.

The sum of the first peaks in each set of gaussians results in a peak with a position very similar to what was defined as the 0-0 peak in the first iteration of the manuscript, i.e., the lowest energy feature in the spectrum at ~ 1.98 eV. The same applies for the 0-1 peak. This is shown in Figure R2 below:

Figure R2: Demonstration of how summing the first and second peaks of the first two sets of gaussians results in peaks with positions very close to those defined as the 0-0 and 0-1 peaks in the first iteration of the manuscript.

The same applies for the g100% absorption spectrum in chlorobenzene, as is shown in Figure R3 and R4 below:

Figure R3: Gaussian fitting of the g100% absorption spectrum in chlorobenzene. The experimental data (black crosses) were fit with the sum of two sets of gaussian peaks, each with a unique, fixed spacing corresponding to a mode in the steady-state Raman spectra.

Figure R4: Demonstration of how summing the first and second peaks of the first two sets of gaussians results in peaks with positions very close to those defined as the 0-0 and 0-1 peaks in the first iteration of the manuscript.

We therefore propose that our original analysis of the data, assigning the 0-0 and 0-1 peaks to the two vibronic features in the 1.9-2.2 eV range of the spectrum is sufficient. It is not possible, however, to plot the reduced absorption and emission spectra as requested. The solution PL and absorption spectra were collected at room temperature using a Shimadzu RF-5301PC fluorescence

spectrometer and Shimadzu UV-2550 spectrometer, respectively. These instruments do not have integrating spheres and so it is not possible to correct the raw spectra obtained. We therefore instead show the HR factor values for each solvent normalised to the g0% polymer value in order to show the relative increasing trend with increasing glycol content:

Figure R5: HR factors normalised to the g0% value for each solvent to demonstrate the increasing trend with increasing glycol content. Values are from the original analysis methodology presented in the first iteration of the manuscript.

Figure 2 in the main manuscript has been revised to show the normalised HR factor values (Line 188):

Figure 2: (a) Schematic of the PESs for the ground (S_0) and excited (S_1) states showing the vertical transitions that occur during the electronic transitions, the PES minima displacement (ΔQ), and the reorganisation energies associated with vibrational relaxation to the PES minimum after an electronic transition occurs. E_{g0}^0 is the energy of the ground state molecule at the optimised geometry of the ground state molecule, E_{g1}^1 is the energy of the excited-state molecule at the optimised geometry of the ground state molecule, E_{g1}^0 is the energy of the ES molecule at the optimised geometry of the ES molecule and E_{g1}^1 is the energy of the ground state molecule at the optimised geometry of the ES molecule. λ_1 and λ_2 are the reorganisation energies associated with IVR on the S_1 and S_0 PESs, respectively. (b) DFT-calculated singlet λ values for a monomer, trimer and pentamer of the g0% and g100% polymers. (c) Stoke's shift values (top) taken as the energy difference between the positions of the 0-0 peak of the solution absorption and PL spectra. Huang-Rhys Factors (bottom) calculated as the ratio of the 1-0/0-0 solution absorption peak intensities, normalised to the g0% value for each solvent to demonstrate the increasing trend with glycol content.

The SI has been revised to describe the above fittings (Line 358):

For HR-factor and Stoke's shift calculations, spectra were deconvoluted into a sum of Gaussian peaks of equal width and spacing in order to identify and extract the positions and intensities of the 0-0 and 0-1 vibronic peaks. We account for there being specific vibrational modes coupling to the optical transition. These are the 1380 cm^{-1} and 1520 cm^{-1} modes, corresponding to peak 1 and peak 4 in the steady-state Raman spectra, respectively. Peaks 1 and 4 show the largest changes upon increasing glycol content and also show large ground-state bleaching in the FSRS spectra, therefore contributing most to the structural change upon photoexcitation as expected. We therefore fit the spectra with a sum of two sets of gaussian peaks, with the peak spacing in one set corresponding to the energy of the 1380 cm^{-1} mode (0.17 eV) and the peak spacing in the other set corresponding to the energy of the 1520 cm^{-1} mode (0.19 eV). Examples for the g0% and g100% absorption spectra in chlorobenzene are shown in Figures S31-S34 below:

Figure S31: Gaussian fitting of the g0% solution absorption spectrum in chlorobenzene. The experimental data (black crosses) were fit with the sum of two sets of gaussian peaks, each with a unique, fixed spacing corresponding to a mode in the steady-state Raman spectra.

The sum of the first peaks in each set of gaussians results in a peak with a position very similar to the lowest energy feature in the spectrum at ~ 1.98 eV. The same applies for the 0-1 peak. This is shown in Figure S32 below:

Figure S32: Demonstration of how summing the first and second peaks of the two sets of gaussians results in peaks with positions very close to the two vibronic features in the 1.9 – 2.2 eV range of the raw data.

We thus assign the 0-0 and 0-1 peaks to the two vibronic features in the 1.9 – 2.2 eV range of the raw spectrum. The same applies for the g100% spectrum:

Figure S33: Gaussian fitting of the g100% absorption spectrum in chlorobenzene. The experimental data (black crosses) were fit with the sum of two sets of gaussian peaks, each with a unique, fixed spacing corresponding to a mode in the steady-state Raman spectra.

Figure S34: S34 Demonstration of how summing the first and second peaks of the two sets of gaussians results in peaks with positions very close to the two vibronic features in the 1.9 – 2.2 eV range of the raw data.

HR factors are calculated and normalised to the g0% value for each solvent to show the increasing trend with glycol content.

In actuality, for a more proper and thorough Franck-Condon analysis, additional temperature dependent PL and absorption spectra would be required,¹⁻⁴ which is beyond the scope of this work. Performing the measurements at room temperature will result in thermal broadening of peaks causing peak FWHM to vary. There will also be other modes contributing that would otherwise be frozen out at low temperature. Despite the above, we believe that the analysis presented here is sufficient. The uncertainty associated with the room temperature measurements is consistent throughout all of the samples and our results give a good estimate and indication of the relative HR factor values. We identify the specific vibrational modes that couple to the optical transition and contribute most to the reorganisation energy, which is backed up by the steady-state Raman and FSRS measurements.

Regarding point 2 concerning the presentation of figure 6. Here, the authors further comments helped to clarify my questions. If I understand correctly, the authors place the PP state at lower energy than the S₁ state as they consider the delocalisation in an excimer (and the associated stabilization) overcompensates the destabilization resulting from a larger spacing. This is possible, yet it also raises the question whether the difference in PP formation between the glycolated and non-glycolated polymer may rather be a result from the different morphologies and associated excimer content than a result from the differences in reorganization energy (the experimentally derived reorganization energies differ by only 10 meV, i.e. less than kT, which makes it hard to believe that this should make much difference). A critical evaluation of this aspect, e.g. in the discussion, might be worthwhile.

I do not wish to render publication of this manuscript difficult. I have detailed here my concerns so that the authors have a chance to consider them, assuming they also wish for their work to be presented in a way accessible and convincing to many readers.

Our response: From our data, we conclude that increasing polymer glycol content increases the total reorganisation energy upon excited-state formation. Computational calculations and solvatochromic measurements show that glycolating the polymer induces a larger inner reorganisation energy. Steady-state analysis reveals that in thin films, increasing glycol sidechain content results in more disordered packing and a more localised π -electron density to within the CPDT unit, which will contribute to a larger outer reorganisation energy. So, the reviewer is correct in saying that morphology will impact PP formation in the thin films. A larger reorganisation energy leads to longer vibrational cooling times and enhances coupling to the PP state. In order to explain these observations and the timescales on which they occur, we propose the schematic in Figure 6 with the PP state minimum lower in energy. The PP state described here is not a hot state as it has undergone vibrational cooling and it is more stable relative to the singlet exciton state. The spatially delocalised, interchain nature of the state means that the minimum of the PP state is below the minimum of the S₁ state in Figure 6.⁵ As the polymers we discuss display disordered packing structures, it is unlikely that excimer species are forming. An excimer is defined as an interchain excitation whereby two identical repeat units on neighbouring chains form a dimer with the excitation spread over the two units. The formation of an excimer requires very closely packed backbones and specific molecular alignment in order to allow for delocalisation of the wavefunction between the two chains.⁶ Here, we are dealing with polymers that do not show very ordered or close packing in the thin film, as is shown by GIWAXS data in Figure 1 and Table S3. So, we do not consider excimers to be forming in these thin film systems and describe the excited-state species as PP states.

Reviewer #5 (Remarks to the Author):

I co-reviewed this manuscript with one of the reviewers who provided the listed reports as part of the Nature Communications initiative to facilitate training in peer review and appropriate recognition for co-reviewers.

References

1. Gierschner, J., Mack, H. G., Lüer, L. & Oelkrug, D. Fluorescence and absorption spectra of oligophenylenevinylenes: Vibronic coupling, band shapes, and solvatochromism. *J. Chem. Phys.* **116**, 8596 (2002).
2. Kroh, D. *et al.* Identifying the Signatures of Intermolecular Interactions in Blends of PM6 with Y6 and N4 Using Absorption Spectroscopy. *Adv. Funct. Mater.* **32**, 2205711 (2022).
3. Brown, P. J. *et al.* Effect of interchain interactions on the absorption and emission of poly(3-hexylthiophene). *Phys. Rev. B* **67**, 064203 (2003).
4. Ho, P. K. H., Kim, J. S., Tessler, N. & Friend, R. H. Photoluminescence of poly(p-phenylenevinylene)–silica nanocomposites: Evidence for dual emission by Franck–Condon analysis. *J. Chem. Phys.* **115**, 2709–2720 (2001).
5. Reid, O. G., Pensack, R. D., Song, Y., Scholes, G. D. & Rumbles, G. Charge photogeneration in neat conjugated polymers. *Chem. Mater.* **26**, 561–575 (2014).
6. Vollbrecht, J. Excimers in organic electronics. *New J. Chem.* **42**, 11249–11254 (2018).

REVIEWER COMMENTS

Reviewer #4 (Remarks to the Author):

The manuscript argues that glycole side chains lead to a higher reorganization energy. In support of this argument, the authors show transient absorption spectra and fs stimulated Raman spectra, Calculations of the reorganization energy and Stokes' shift, as well as steady state spectra of absorption and emission (SI, Fig S30).

I agree with the authors that a Franck-Condon analysis is not the focus of the work. It is merely useful as a tool check if the trends calculated for reorganization energy and Stokes' shift are consistent with the experimental data. I am confused by the presented FC-analysis, as the absorption data is fitted by two independent progressions with different 0-0 peaks, rather than as one progression with two modes and only one 0-0 peak. Also, the PL (or at least the 0-0 peak in the PL) is not fitted, which makes it difficult to evaluate any trends in the Stokes' shift upon glycation. Perhaps the authors could look into this? I do not want to get hung up on this, as the Raman spectra indeed indicate a stronger pi-electron localization upon glycolation, yet a good consistency between the calculations and the experimental data would be reassuring.

Regarding the relative positions of the minima in the S1 state and the PP state, I accept their reasoning that the GIWAX data do not show any long-range structure. The question then remains, however, how the energy of a state characterized by a larger electron-hole distance (and hence lower binding energy) can be lower than the more tightly bound state (with higher binding energy) from which it originates. Could the authors clarify this?

Reviewer #5 (Remarks to the Author):

Slow Vibrational Relaxation Drives Ultrafast Formation of Photoexcited Polaron Pair States in Glycolated Conjugated Polymers

Manuscript: NCOMMS-23-34380A

Responses to the reviewers' comments

We thank all reviewers for their time to read the manuscript and for their encouraging and constructive comments. We believe that we have addressed all of the reviewers' comments. Below are our detailed replies to the reviewers' comments and questions. Our detailed point by point responses are in blue and the changes in the manuscript are highlighted in green. We hope that our revised manuscript is now acceptable for publication in *Nature Communications*.

Reviewer #4 (Remarks to the Author):

The manuscript argues that glycole side chains lead to a higher reorganization energy. In support of this argument, the authors show transient absorption spectra and fs stimulated Raman spectra, Calculations of the reorganization energy and Stokes' shift, as well as steady state spectra of absorption and emission (SI, Fig S30).

I agree with the authors that a Franck-Condon analysis is not the focus of the work. It is merely useful as a tool check if the trends calculated for reorganization energy and Stokes' shift are consistent with the experimental data. I am confused by the presented FC-analysis, as the absorption data is fitted by two independent progressions with different 0-0 peaks, rather than as one progression with two modes and only one 0-0 peak. Also, the PL (or at least the 0-0 peak in the PL) is not fitted, which makes it difficult to evaluate any trends in the Stokes' shift upon glycation. Perhaps the authors could look into this? I do not want to get hung up on this, as the Raman spectra indeed indicate a stronger pi-electron localization upon glycolation, yet a good consistency between the calculations and the experimental data would be reassuring.

Regarding the relative positions of the minima in the S1 state and the PP state, I accept their reasoning that the GIWAX data do not show any long-range structure. The question then remains, however, how the energy of a state characterized by a larger electron-hole distance (and hence lower binding energy) can be lower than the more tightly bound state (with higher binding energy) from which it originates. Could the authors clarify this?

Our response: We thank the reviewer for comments. To extract Stokes' Shift values from the solution absorption and PL spectra, we have taken the definition of Stokes' Shift to be the energy difference between the absorption and emission maxima.^{1,2} We have corrected the data to show the reduced absorption and emission. The corrected, normalised data is shown in Figure R1 below:

Figure R1: Normalised absorbance and PL spectra (in photons per energy interval) of the g0% (top), g50% (middle) and g100% (bottom) polymers in solution. Spectra were taken in toluene, chlorobenzene (CB), chloroform (CF) and dichloromethane (DCM), and polymers were at a concentration of 0.01 mg/ml. A shoulder of emission can be seen in the g0% DCM absorption spectrum, which likely arises due to the poor solubility of the alkylated polymers in the polar solvent.

The SI has been revised accordingly (Line 354):

Figure S30: Normalised absorbance and PL spectra (in photons per energy interval) of the g0% (top), g50% (middle) and g100% (bottom) polymers in solution. Spectra were taken in toluene, chlorobenzene (CB), chloroform (CF) and dichloromethane (DCM), and polymers were at a concentration of 0.01 mg/ml. A shoulder of emission can be seen in the g0% DCM absorption spectrum, which likely arises due to the poor solubility of the alkylated polymers in the polar solvent.

We have re-extracted the Stokes' shift and have made the extraction method clearer in Figure R2 and Table R1 shown below:

Figure R2: Method of Stokes' shift extraction. The Stokes' shift is taken as the energy difference between the maxima of the absorbance and PL spectra, which are indicated with vertical dashed lines for each spectrum. An example of Stokes' shift extractions is shown for the toluene spectra (top).

Table R1: Maxima positions of the absorbance and PL spectra for each polymer and the associated Stokes' shifts.

Polymer	Solvent	Absorbance maximum (eV)	PL maximum (eV)	Stokes' shift (eV)
g0%	Toluene	1.996	1.870	0.126
	CB	1.977	1.845	0.132
	CF	1.984	1.853	0.131
	DCM	1.987	1.864	0.123
g50%	Toluene	2.009	1.881	0.128
	CB	1.990	1.853	0.137
	CF	2.000	1.859	0.141
	DCM	1.996	1.864	0.130
g100%	Toluene	2.029	1.884	0.145
	CB	2.006	1.864	0.142
	CF	2.019	1.867	0.152
	DCM	2.019	1.876	0.144

Regarding the Franck-Condon fittings to the data, we have attempted to re-fit the data with one progression and two phonon modes, with poor results that did not allow for reliable extraction of the HR factors. In actuality, for a more proper and thorough Franck-Condon analysis, additional temperature dependent PL and absorption spectra would be required,³⁻⁶ which is beyond the scope of this work. As the referee agreed, the Franck-Condon analysis is not the focus of the work. Performing the measurements at room temperature will result in thermal broadening of peaks causing peak FWHM to vary. There will also be other modes contributing that would otherwise be frozen out at low temperature. There may also still be some aggregation, despite a low concentration being used. We therefore propose to remove extraction of the HR factors from Figure 2, keeping the Stokes' shift to support the conclusions made from the DFT calculations.

Figure 2 (Line 188) and the SI (Line 359) have been revised to include these changes:

Figure 2: (a) Schematic of the PESs for the ground (S_0) and excited (S_1) states showing the vertical transitions that occur during the electronic transitions, the PES minima displacement (ΔQ), and the reorganisation energies associated with vibrational relaxation to the PES minimum after an electronic transition occurs. E_{g0}^0 is the energy of the ground state molecule at the optimised geometry of the ground state molecule, E_{g0}^1 is the energy of the excited-state molecule at the optimised geometry of the ground state molecule, E_{g1}^1 is the energy of the ES molecule at the optimised geometry of the ES molecule and E_{g1}^0 is the energy of the ground state molecule at the optimised geometry of the ES molecule. λ_1 and λ_2 are the reorganisation energies associated with IVR on the S_1 and S_0 PESs, respectively. (b) DFT-calculated singlet λ values for a monomer, trimer and pentamer of the $g0\%$ and $g100\%$ polymers. (c) Stokes shift values taken as the energy difference between the positions of the solution absorption and PL spectra maxima.

Figure S31: Method of Stokes' shift extraction. The Stokes' shift is taken as the energy difference between the maxima of the absorbance and PL spectra, which are indicated with vertical dashed lines for each spectrum. An example of Stokes' shift extractions is shown for the toluene spectra (top).

Table S4: Maxima positions of the absorbance and PL spectra for each polymer and the associated Stokes' shifts.

Polymer	Solvent	Absorbance maximum (eV)	PL maximum (eV)	Stokes' shift (eV)
g0%	Toluene	1.996	1.870	0.126
	CB	1.977	1.845	0.132
	CF	1.984	1.853	0.131
	DCM	1.987	1.864	0.123
g50%	Toluene	2.009	1.881	0.128
	CB	1.990	1.853	0.137
	CF	2.000	1.859	0.141
	DCM	1.996	1.864	0.130
g100%	Toluene	2.029	1.884	0.145
	CB	2.006	1.864	0.142
	CF	2.019	1.867	0.152

The reviewer also asked about the relative positions of the minima of the singlet state and the polaron pair state in our schematic diagram in Figure 6. We proposed a schematic diagram with the polaron pair state minimum lying lower in energy than the singlet state. We define the polaron pair state to be an exciton with 'increased charge transfer, with the electron-hole pair being spatially separated into different polymer chains but still bound with a Coulomb attraction.' Importantly, the polaron pairs are not fully charge separated states, the electron and hole are still Coulombically bound. This is evidenced by the fluence independence of the TAS data (Figure S39). Additionally, the population of charges at later timescales (μs) is very low (Figure S37). The polaron pair state is predominantly intermolecular in nature, being delocalised between neighbouring polymer chains.⁷ The spatially delocalised, interchain nature of the state is reflected in the minimum of the polaron pair state being below the minimum of the S_1 state. This is equivalent to a bound CT-state commonly found in a donor-acceptor blend system, which is stabilised relative to the S_1 state.⁸

The polaron pairs can have a larger relaxation energy compared to the excitons, which can lower the polaron pair state energy minimum. In addition, the polaron pairs can have a larger external environment impact (e.g. the larger external reorganisation energy) due to their charged nature compared to the excitons that are neutral. Additionally, if the polaron pair state were always higher in energy than the singlet exciton state, free charge generation would be difficult, which is not always true.⁹⁻¹²

References

1. Gierschner, J., Cornil, J. & Egelhaaf, H.-J. Optical Bandgaps of π -Conjugated Organic Materials at the Polymer Limit: Experiment and Theory. *Adv. Mater.* **19**, 173–191 (2007).
2. Nijegorodov, N. ., Downey, W. . & Danailov, M. . Systematic investigation of absorption, fluorescence and laser properties of some p- and m-oligophenylenes. *Spectrochim. Acta Part A Mol. Biomol. Spectrosc.* **56**, 783–795 (2000).
3. Gierschner, J., Mack, H. G., Lüer, L. & Oelkrug, D. Fluorescence and absorption spectra of oligophenylenevinyls: Vibronic coupling, band shapes, and solvatochromism. *J. Chem. Phys.* **116**, 8596 (2002).
4. Kroh, D. *et al.* Identifying the Signatures of Intermolecular Interactions in Blends of PM6 with Y6 and N4 Using Absorption Spectroscopy. *Adv. Funct. Mater.* **32**, 2205711 (2022).
5. Brown, P. J. *et al.* Effect of interchain interactions on the absorption and emission of poly(3-hexylthiophene). *Phys. Rev. B* **67**, 064203 (2003).
6. Ho, P. K. H., Kim, J. S., Tessler, N. & Friend, R. H. Photoluminescence of poly(p-phenylenevinylene)–silica nanocomposites: Evidence for dual emission by Franck–Condon analysis. *J. Chem. Phys.* **115**, 2709–2720 (2001).
7. Reid, O. G., Pensack, R. D., Song, Y., Scholes, G. D. & Rumbles, G. Charge photogeneration in neat conjugated polymers. *Chem. Mater.* **26**, 561–575 (2014).
8. Clarke, T. M. & Durrant, J. R. Charge photogeneration in organic solar cells. *Chem. Rev.* **110**, 6736–6767 (2010).
9. Durrant, J. *et al.* Octupole Moment Driven Free Charge Generation in Partially Chlorinated

Subphthalocyanine for Planar Heterojunction Organic Photodetectors. (2023).
doi:10.21203/RS.3.RS-3324499/V1

10. Price, M. B. *et al.* Free charge photogeneration in a single component high photovoltaic efficiency organic semiconductor. *Nat. Commun.* **13**, 2827 (2022).
11. Fu, Y. *et al.* Molecular orientation-dependent energetic shifts in solution-processed non-fullerene acceptors and their impact on organic photovoltaic performance. *Nat. Commun.* **14**, 1870 (2023).
12. Park, S. Y. *et al.* The State-of-the-Art Solution-Processed Single Component Organic Photodetectors Achieved by Strong Quenching of Intermolecular Emissive State and High Quadrupole Moment in Non-Fullerene Acceptors. *Adv. Mater.* **35**, 2306655 (2023).

REVIEWERS' COMMENTS

Reviewer #4 (Remarks to the Author):

The revised version has clarified many of the issues. Regarding the energy of the polaron pair relative to the singlet excited state, I still argue that in a homogeneous, one-component system, the state formed by a pair of charges separated by some distance will always be higher in energy than that of the charge pair at close distance, where they are more tightly bound. This is different from the case of a charge-transfer state in a donor-acceptor system, and it is also different in a system made up by one material if the material prevails in two different phases, e.g. amorphous and partially crystallized. It is conceivable that such morphological homogeneities prevail in the samples investigated, which would lower the energy of the pair state relative to the singlet, even if they are too small to be picked up by structural techniques such as GIWAXS. A comment on this in the manuscript might be useful. I feel the manuscript is suitable for publication without further review.

Slow vibrational relaxation drives ultrafast formation of photoexcited polaron pair states in glycolated conjugated polymers

Manuscript: NCOMMS-23-34380A

Responses to the reviewers' comments

We thank all reviewers for their time to read the manuscript and for their encouraging and constructive comments. We believe that we have addressed all of the reviewers' comments. Below are our detailed replies to the reviewers' comments and questions. Our detailed point by point responses are in blue and the changes in the manuscript are highlighted in magenta.

Reviewer #4 (Remarks to the Author):

The revised version has clarified many of the issues. Regarding the energy of the polaron pair relative to the singlet excited state, I still argue that in a homogeneous, one-component system, the state formed by a pair of charges separated by some distance will always be higher in energy than that of the charge pair at close distance, where they are more tightly bound. This is different from the case of a charge-transfer state in a donor-acceptor system, and it is also different in a system made up by one material if the material prevails in two different phases, e.g. amorphous and partially crystallized. It is conceivable that such morphological homogeneities prevail in the samples investigated, which would lower the energy of the pair state relative to the singlet, even if they are too small to be picked up by structural techniques such as GIWAXS. A comment on this in the manuscript might be useful.

I feel the manuscript is suitable for publication without further review.

Our response: We thank the reviewer for their comments. We acknowledge their point regarding the relative energies of the singlet and PP states and have added a sentence in the manuscript to bring attention to this point (Line 378):

'As this is a thin film homopolymer system, the PP state is likely predominantly intermolecular in nature, being delocalised between packed polymer chains.⁵⁴ Given the disordered packing nature of both polymers in the thin film, it is likely that the e-h pair of the PP will be situated on polymer chains in different phases, for example amorphous and crystallised regions. This stabilises the PP state relative to the singlet exciton, similar to how a CT-state in a donor-acceptor blend system is stabilised relative to the singlet state.^{1'}